# EQUIVARIANT GRAPH NEURAL OPERATOR FOR MODELING 3D DYNAMICS

## ABSTRACT

Modeling the complex three-dimensional (3D) dynamics of relational systems is an important problem in the natural sciences, with applications ranging from molecular simulations to particle mechanics. Machine learning methods have achieved good success by learning graph neural networks to model spatial interactions. However, these approaches do not faithfully capture temporal correlations since they only model next-step predictions. In this work, we propose Equivariant Graph Neural Operator (EGNO), a novel and principled method that directly models dynamics as trajectories instead of just as next-step prediction. Different from existing methods, EGNO explicitly learns the temporal evolution of 3D dynamics where we formulate the dynamics as a function over time and learn neural operators to approximate it. To capture the temporal correlations while keeping the intrinsic SE(3)-equivariance, we develop equivariant temporal convolutions parameterized in the Fourier space and build EGNO by stacking the Fourier layers over equivariant networks. Comprehensive experiments in multiple domains, including particle simulations, human motion capture, and molecular dynamics, demonstrate the significantly superior performance of EGNO against existing methods, thanks to the equivariant temporal modeling.

## 1 INTRODUCTION

Modeling the dynamics of many body systems in Euclidean space is an important problem in machine learning (ML) at all scales (Tenenbaum et al., 2011; Battaglia et al., 2016), ranging from interactions of atoms in a molecule or protein to interactions of celestial bodies in the universe. Traditional methods tackle these physical dynamics by Newton's laws along with interactions calculated from physical rules, *e.g.*, Coulomb force for modeling systems with charged particles (Kipf et al., 2018). Such calculations based on physical rules typically are very computationally expensive and slow, and recently researchers are studying alternative ML approaches by learning neural networks (NNs) to automatically capture the physical rules and model the interactions in a data-driven fashion. Along this research direction, considerable progress has been achieved in recent years by regarding objects as nodes and interactions as edges and learning graphs neural networks (GNNs) and learn the spatial interactions via message-passing schemes. (Battaglia et al., 2016; Kipf et al., 2018; Sanchez-Gonzalez et al., 2019; Martinkus et al., 2021; Sanchez-Gonzalez et al., 2020; Pfaff et al., 2020).

Among the advancements, Equivariant GNN (EGNN) is a state-of-the-art approach with impressive results in modeling physical dynamics in 3D Euclidean space (Thomas et al., 2018; Satorras et al., 2021; Huang et al., 2022). EGNNs possess equivariance to roto-translational transformations in the Euclidean space, which has been demonstrated as a vital inductive bias to improve generalization (Köhler et al., 2019; Fuchs et al., 2020; Han et al., 2022; Xu et al., 2022). However, these methods typically model the dynamics by only learning to predict the next state given the current state, failing to faithfully capture the temporal correlation along the trajectory. As a result, performance is still often unsatisfactory without fully understanding the dynamics as systems evolve through time.

**Our contributions:** In this paper, we propose Equivariant Graph Neural Operator (EGNO), a novel and principled method to overcome the above challenge by directly modeling the entire trajectory dynamics instead of just the next time-step prediction. Different from existing approaches, EGNO predicts dynamics as a temporal function that is not limited to a fixed discretization. Our framework is inspired by the neural operator (NO) (Li et al., 2020; Kovachki et al., 2021b;a), specifically, Fourier

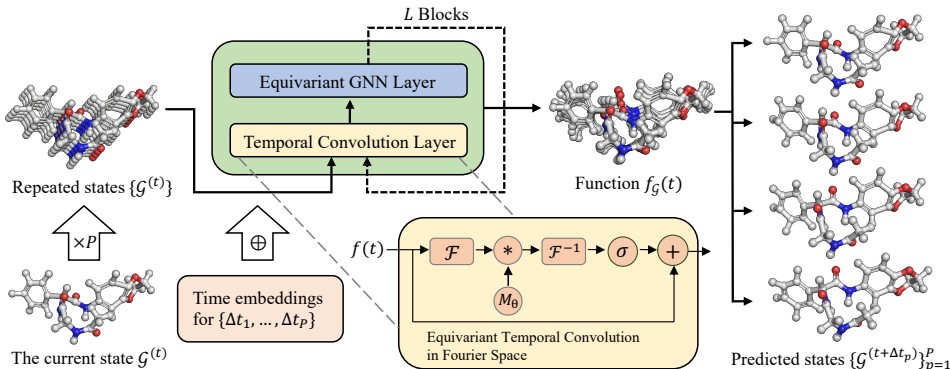

Figure 1: Illustration of EGNO. EGNO blocks (green) can be built with any EGNN layers (blue) and the proposed equivariant temporal convolution layers (yellow). Consider discretizing the time window $\Delta T$ into $P$ points $\{\Delta t_1, \ldots, \Delta t_P\}$. Given a current state $\mathcal{G}^{(t)}$, we will first repeat its features by $P$ times, concatenate the repeated features with time embeddings, and feed them into $L$ EGNO blocks. Within each block, the temporal layers operate on temporal and channel dimensions while the EGNN layers operate on node and channel dimensions. Finally, EGNO can predict future dynamics as a function $f_{\mathcal{G}}(t)$ and decode a trajectory of states $\{\mathcal{G}^{(t+\Delta t_p)}\}_{p=1}^{P}$ in parallel.

neural operator (FNO) (Li et al., 2021; Zheng et al., 2023), which has shown great effectiveness in learning maps between function spaces with desirable discretization convergence guarantees. Our core idea is to formulate the physical dynamics as a function over time and learn neural operators to approximate it. The main challenge in developing EGNO is to capture the temporal correlations while still keeping the SE(3)-equivariance in the Euclidean space. To this end, we develop equivariant temporal convolution layers in the Fourier space and realize EGNO by stacking them with equivariant networks, as shown in Figure 1. Our key innovation is that we notice the equivariance property of Fourier and inverse Fourier transforms and keep this equivariance in Fourier space with special kernel integral operators. These designs result in an efficient operator learning framework that is capable of mapping a current state directly to the solution trajectories, while retaining 3D spatial equivariance.

EGNO enjoys several unique advantages compared to existing GNN-based methods, thanks to the proposed equivariant operator learning framework. Firstly, our method explicitly learns to model the trajectory while still keeping the intrinsic symmetries in Euclidean space. This, in practice, leads to more expressive modeling of underlying dynamics and achieves higher state prediction accuracy. Secondly, our operator formulation enables efficient parallel decoding of future states (within a time window) with just one model inference, and the model is not limited to one fixed temporal discretization. This allows users to run dynamics inference at any timestep size without switching model parameters. Finally, our proposed temporal convolutional layer is very general and can be easily combined with any specially designed EGNN layers. This permits EGNO to be easily deployed in a wide range of different physical dynamics scenarios.

We conduct comprehensive experiments on multiple benchmarks to verify the effectiveness of the proposed method, including particle simulations, motion captures, and molecular dynamics for both small molecules and large proteins. Experimental results show that EGNO can consistently achieve superior performance over existing methods by a significant margin on various datasets. For instance, EGNO incurs a relative improvement of 36% over EGNN for Aspirin molecular dynamics, and 52% on average for human motion capture. All the empirical studies suggest that EGNO enjoys a higher capacity to model the geometric dynamics, thanks to the equivariant temporal modeling.

## 2 RELATED WORK

### 2.1 GRAPH NEURAL NETWORKS

Graph neural networks (Gilmer et al., 2017; Kipf & Welling, 2017) are a family of neural network architectures for representation learning on relational structures, which has been widely adopted in modeling complex interactions and simulating physical dynamics. Early research in this field improves performance by designing expressive modules to capture the system interactions (Battaglia

et al., 2016; Kipf et al., 2018; Mrowca et al., 2018) or imposing physical mechanics (Sanchez-Gonzalez et al., 2019). Beyond interactions, another crucial consideration for simulating physical dynamics is the symmetry in Euclidean space, *i.e.*, the equivalence *w.r.t* rotations and translations. To this end, several works are first proposed to enforce translation equivariance (Ummenhofer et al., 2019; Sanchez-Gonzalez et al., 2020; Pfaff et al., 2020; Wu et al., 2018), and then later works further enforce the rotation equivariance by spherical harmonics (Thomas et al., 2018; Fuchs et al., 2020) or equivariant message passings (Satorras et al., 2021; Huang et al., 2022). In addition, some methods (Du et al., 2022; Morehead & Cheng, 2023) also introduce architectural enhancements with local frame construction to process geometric features while satisfying equivariance. However, despite the considerable progress, all the models are limited to modeling only the spatial interactions and ignore the critical temporal correlations, while in this paper, we view dynamics as a function of geometric states over time and explicitly learn the temporal correlations.

## 2.2 NEURAL OPERATORS

Neural operators (Kovachki et al., 2021b) are machine learning paradigms for learning mappings between Banach spaces, *i.e.*, continuous function spaces (Li et al., 2020; Kovachki et al., 2021a), which have been widely adopted as data-driven surrogates for solving partial differential equations (PDE) or ordinary differential equations (ODE) in scientific problems. Among them, Fourier neural operator (Li et al., 2021) is one of the state-of-the-art methods for solving PDEs across many scientific problems (Yang et al., 2021; Wen et al., 2022). It holds several vital properties including discretization invariance and approximation universality (Kovachki et al., 2021a;b), which is useful in our scenario where we aim to model the dynamics as a function of 3D states over time. However, previous studies mainly concentrate on functions defined on spatial domains (Yang et al., 2021; Wen et al., 2022) or temporal functions with simple scalar outputs (Zheng et al., 2023). By contrast, in this paper, we are interested in modeling geometric dynamics with temporal functions describing the evolution of directional features where it is nontrivial to enforce the symmetry of Euclidean space.

## 3 PRELIMINARIES

### 3.1 NOTIONS AND NOTATIONS

In this paper, we consider modeling the dynamics of multi-body systems, *i.e.*, a sequence of geometric graphs $\mathcal{G}^{(t)}$ indexed by time $t$. It can also be viewed as a function of 3D states over time $f_{\mathcal{G}} : t \to \mathcal{G}^{(t)}$. Suppose we have $N$ particles or nodes in the system, then the graph $\mathcal{G}$ at each timestep can be represented as the feature map $[\mathbf{h}, \mathbf{Z}]$, where $\mathbf{h} = (\mathrm{h}_1, \ldots, \mathrm{h}_N) \in \mathbb{R}^{N \times k}$ is the node feature matrix of $k$ dimension (*e.g.*, atom types and charges), and $\mathbf{Z} \in \mathbb{R}^{N \times m \times 3}$ is the generalized matrix composed of $m$ directional tensors in 3D. For example, $\mathbf{Z}$ can be instantiated as the concatenation of coordinate matrix $\mathbf{x} \in \mathbb{R}^{N \times 3}$ and velocity matrix $\mathbf{v} \in \mathbb{R}^{N \times 3}$, leading to $\mathbf{Z} = [\mathbf{x}, \mathbf{v}] \in \mathbb{R}^{N \times 2 \times 3}$. Specifically, in this paper, we consider the dynamics where $\mathbf{h}$ will stay unaffected *w.r.t.* time while $\mathbf{Z}$ is updated, *e.g.*, for molecular dynamics, atom types will remain unchanged while positions are repeatedly updated.

### 3.2 EQUIVARIANCE

*Equivariance* is important and ubiquitous in 3D systems (Thomas et al., 2018; Fuchs et al., 2020). Let $g \in G$ denote an abstract group, then a function $f : X \to Y$ is defined as equivariant *w.r.t* $g$ if there exists equivalent transformations $T_g$ and $S_g$ for $g$ such that $f \circ T_g(\mathbf{Z}) = S_g \circ f(\mathbf{Z})$ (Serre et al., 1977). In this paper, we consider the Special Euclidean group SE(3) for modeling the 3D dynamics, which is the group of 3D rotations and translations. Then transformations $T_g$ and $S_g$ can be represented by a translation $\boldsymbol{\mu} \in \mathbb{R}^3$ and an orthogonal rotation matrix $\mathbf{R} \in \mathrm{SO}(3)$.

As a practical example, to model dynamics by learning a function to predict $\mathcal{G}^{(t+1)}$ from $\mathcal{G}^{(t)}$, we hope that any rotations and translations applied to the current state coordinate $\mathbf{x}^{(t)}$ should be accordingly applied on the predicted next one $\mathbf{x}^{(t+1)}$, while the velocities $\mathbf{v}$ should also rotate in the same way but remain unaffected by translations[1]. Such inductive bias has been shown critical in improving data efficiency and generalization for geometric modeling (Cohen et al., 2018; Xu et al., 2023).

---

[1]Following this convention, in this paper we use $\mathbf{R}\mathbf{Z} + \boldsymbol{\mu}$ as the shorthand for $[\mathbf{R}\mathbf{x} + \boldsymbol{\mu}, \mathbf{R}\mathbf{v}]^{\mathrm{T}}$.

### 3.3 FOURIER NEURAL OPERATOR

Fourier neural operator (FNO) (Li et al., 2021) is an advanced data-driven approach for learning mappings between Banach spaces, which has been widely deployed in solving PDEs and ODEs (Kovachki et al., 2021b;a; Zheng et al., 2023) in scientific problems. Typically, a Fourier neural operator $F$ can be implemented as a stack of $L$ kernel integration layers with each kernel function parameterized by learnable weights $\theta$. Let $\sigma$ denote nonlinear activation functions, we have

$$F_\theta := Q \circ \sigma(W_L + \mathcal{K}_L) \circ \cdots \circ \sigma(W_1 + \mathcal{K}_1) \circ P \tag{1}$$

where $P$ and $Q$ are lifting and projection operators parameterized by neural networks, which lift inputs to higher channel space and finally project them back to the target domain, respectively. $W_i$ are point-wise linear transformations, and $\mathcal{K}_i$ are kernel integral operators parameterized in Fourier space. Formally, given $f_i$ as the input function to the $i$-th layer, $\mathcal{K}_i$ is defined as:

$$(\mathcal{K}_i f)(t) = \mathcal{F}^{-1}(\mathbf{M}_i \cdot (\mathcal{F}f))(t), \quad \forall t \in D, \tag{2}$$

where $\mathcal{F}$ and $\mathcal{F}^{-1}$ denote the Fourier and inverse Fourier transforms on temporal domain $D$, respectively, and $\mathbf{M}_i$ is the learnable parameters that parameterizes $\mathcal{K}_i$ in Fourier space. Starting from input function $a(t)$, we first lift it to higher dimension space by $P$, apply $L$ layers of integral operators and activation functions, and then project back to the target dimension with $Q$. The resulting FNO is shown to follow the critical discretization invariance of the function domain $D$.

## 4 EQUIVARIANT GRAPH NEURAL OPERATOR

### 4.1 OVERVIEW

Our goal is to learn a neural operator that given the current structure $\mathcal{G}^{(t)}$ (recurrently) predicts the future structures $\mathcal{G}^{(t+\Delta t)}$ with $\Delta t > 0$. Previous methods (Thomas et al., 2018; Gilmer et al., 2017; Wang & Chodera, 2023) mainly tackled the problem by learning to predict $\mathcal{G}^{(t+1)}$ from $\mathcal{G}^{(t)}$, while in this paper we instead learn to predict a future trajectory $\{\mathcal{G}^{(t+\Delta t)} : \Delta t \sim D = [0, \Delta T]\}$ within a time window $\Delta T$. Let $\mathcal{U} : D \to \mathbb{R}^{N \times m \times 3}$ denote the space of the target temporal functions that predict future states $\mathbf{Z}^{(t+\Delta t)}$. Assume $F^\dagger$ to be the solution operator, we can define EGNO as $F_\theta$ and learn it with the following objective function:

$$\min_\theta \mathbb{E}_{\mathcal{G}^{(t)} \sim p_{\text{data}}} \mathcal{L}(F_\theta(\mathcal{G}^{(t)})(t) - F^\dagger(\mathcal{G}^{(t)})(t)), \tag{3}$$

where $\mathcal{L} : \mathcal{U} \to \mathbb{R}_+$ is any loss function such as $L_p$-norm. Further, the dynamics introduced in section 3.1 can generally be described by Newtonian equations of motion

$$d\mathbf{Z} = \begin{bmatrix} d\mathbf{x} \\ d\mathbf{v} \end{bmatrix} = \begin{bmatrix} \mathbf{v} \cdot dt \\ -\mathbf{r} \cdot dt - \gamma \mathbf{v} \cdot dt \end{bmatrix}, \tag{4}$$

where $\mathbf{r}$ represents the force of particle interactions, and $\gamma$ is the weight controlling the potential friction term. From equation (4), we know there exists an exact solution of $\mathbf{Z}^{(t)}$ corresponding to the solution operator $F^\dagger$ in equation (3), which represents the solver for solving the dynamics function, i.e., mapping a current structure $\mathcal{G}^{(t)}$ to the function $f_{\mathcal{G}}(\Delta t)$ that describes future states $\mathcal{G}^{(t+\Delta t)}$ for $\Delta t \sim [0, \Delta T]$. According to neural operator theory, we have that the FNO framework can universally approximate the dynamics and predict future geometric structures $\mathcal{G}^{(t+\Delta t)}$ with one forward inference, and we leave the formal statement in Appendix A.1.

### 4.2 EQUIVARIANT TEMPORAL CONVOLUTIONS IN FOURIER SPACE

In this part, we formally introduce our temporal convolution layers $\mathcal{T}_\theta$, which are built upon Fourier integration operators $\mathcal{K}_\theta$ to model the temporal correlations. Let $f : D \to \mathcal{G}^2$ be the input starting function that describes structures $\mathcal{G}^{(t)}$ for time $t$, i.e., $f(t) = [f_\mathbf{h}, f_\mathbf{Z}(t)]^\mathsf{T}$. Then the convolution layer $\mathcal{T}_\theta$ is implemented as:

$$(\mathcal{T}_\theta f)(t) = f(t) + \sigma((\mathcal{K}_\theta f)(t)), \tag{5}$$

---

[2]For simplicity, in Section 4.1 we omit the subscript and use $f$ to denote $f_{\mathcal{G}}$.

which is a modified version of the FNO layer in equation (1). In particular, we move the nonlinearity $\sigma$ and trainable parameters to the integration layer $\mathcal{K}$ and leave the linear transform $W$ as an identical residual connection (He et al., 2016), which has shown effectiveness in various applications including operator learning (Li et al., 2021; Zheng et al., 2023). The temporal integration layer $\mathcal{K}_\theta$ is implemented directly in the frequency domain, following equation (2), where we first conduct Fourier transform over $f$ and then multiply it with $\mathbf{M}_\theta$. In the following, we introduce how we design $\mathcal{K}_\theta$ to ensure efficient temporal modeling while keeping the crucial SE(3)-equivariance.

**Equivariance.** Different from typical FNO operating on non-directional function space, in this work we aim to learn functions $f(t)$ for predicting geometric structures $\mathcal{G}^{(t)}$, where the output is directional in 3D and requires the vital SE(3)-equivariance inductive bias. To be more specific, the convolution layer $\mathcal{T}_\theta$ integrates a sequence of geometric structures as inputs and outputs a new sequence of updated structures, and the equivariance requirement means that the output ones should transform accordingly if a global roto-translation transformation is applied on the input ones. Let $\mathbf{R}$ and $\boldsymbol{\mu}$ denote the rotation matrix and translation vector respectively, then we have

$$(\mathcal{T}_\theta(\mathbf{R}f + \boldsymbol{\mu}))(t) = (\mathbf{R}(\mathcal{T}_\theta f) + \boldsymbol{\mu})(t), \tag{6}$$

where $\mathbf{R}f + \boldsymbol{\mu}$ is the shorthand for $[f_\mathbf{h}, \mathbf{R}f_\mathbf{Z} + f_{\boldsymbol{\mu}}]$. This means with roto-translational transformations, the directional features will be affected while non-directional node features stay unaffected.

However, it is non-trivial to impose such equivariance into FNOs. Our key innovation is to implement $\mathcal{K}_\theta$ with a block diagonal matrix $\mathbf{M}_\theta$ in equation (2). Formally, we have

$$(\mathcal{K}_\theta f)(t) = \mathcal{F}^{-1}(\begin{bmatrix} \mathbf{M}_\theta^\mathbf{h} & \mathbf{0} \\ \mathbf{0} & \mathbf{M}_\theta^\mathbf{Z} \end{bmatrix} \cdot (\mathcal{F} \begin{bmatrix} f_\mathbf{h} \\ f_\mathbf{Z} \end{bmatrix}))(t), \tag{7}$$

where $\mathbf{M}_\theta^\mathbf{h} \in \mathbb{C}^{I \times k \times k}$ and $\mathbf{M}_\theta^\mathbf{Z} \in \mathbb{C}^{I \times m \times m}$ are two complex-valued matrices. The $\mathcal{F}$ and $\mathcal{F}^{-1}$ are realized with fast Fourier transform with $I$ being the hyperparameter to control the maximal number of frequency modes. Then for the frequency domain, we will truncate the modes higher than $I$ and thus have $\mathcal{F}(f_\mathbf{h}) \in \mathbb{C}^{I \times k}$ and $\mathcal{F}(f_\mathbf{Z}) \in \mathbb{C}^{I \times m \times 3}$ respectively[3]. The product of Fourier transform results and kernel functions are given by:

$$[\mathbf{M}_\theta^\mathbf{h} \cdot (\mathcal{F}f_\mathbf{h})]_{i,j} = \sum_{l=1}^{k} (\mathbf{M}_\theta^\mathbf{h})_{i,j,l} (\mathcal{F}f_\mathbf{h})_{i,l} \qquad [\mathbf{M}_\theta^\mathbf{Z} \cdot (\mathcal{F}f_\mathbf{Z})]_{i,s} = \sum_{l=1}^{m} (\mathbf{M}_\theta^\mathbf{Z})_{i,s,l} (\mathcal{F}f_\mathbf{Z})_{i,l} \tag{8}$$

for all $i \in \{1, \ldots, I\}$, $j \in \{1, \ldots, k\}$ and $s \in \{1, \cdots, m\}$. The main difference between the two products lies in that the first is simply scalar multiplications, while the second multiplies scalar with 3D vectors. Furthermore, to ensure equivariance, we implement the nonlinearity in equation (5) by

$$\sigma(\mathcal{K}_\theta f) = [\sigma(\mathcal{K}_\theta f_\mathbf{h}), \mathcal{K}_\theta f_\mathbf{Z}]^\mathrm{T}, \tag{9}$$

which means we skip the activation layer for $\mathbf{Z}$ part and thus keep its original orientations. With the above designs, we enjoy the following crucial property:

**Theorem 4.1.** *By parameterizing the kernel function $\mathcal{K}_\theta$ with equations (7) and (9), we have that $\mathcal{T}_\theta$ is an SO(3)-equivariant operator, i.e., $(\mathcal{T}_\theta(\mathbf{R}f))(t) = (\mathbf{R}(\mathcal{T}_\theta f))(t)$.*

The key foundations behind the theorem are that $\mathcal{F}$ and $\mathcal{F}^{-1}$ are both equivariant operations, and we keep the equivariance in the frequency domain by updating 3D directional vectors with linear combinations. We provide the full proof for the theorem in Appendix A.

**Temporal discretization.** We discretize the time domain and perform discrete Fourier transforms for better computational efficiency. Suppose the time window with length $\Delta T$ is discretized into $P$ points. Considering the domains of the input and output functions of temporal convolution layer $\mathcal{T}_\theta$ are both states $\mathcal{G} \in \mathbb{R}^{N \times (k+m \times 3)}$. Then $f(t)$ can be represented as a sequence of $P$ states of structures, *i.e.*, a tensor in $\mathbb{R}^{P \times N \times (k+m \times 3)}$. Thus, we can perform $\mathcal{F}$ and $\mathcal{F}^{-1}$ in equation (7) efficiently with fast Fourier transform algorithm.

---

[3]Note that, $k$ and $m$ can be viewed as numbers of channels, and the temporal convolutions only operate on the temporal and feature channel dimensions. Therefore, here the node dimension $N$ is just treated as the same as the batch dimension and omitted in the expressions.

### 4.3 GENERALIZED ARCHITECTURE

As shown in Figure 1, the EGNO is very generalized and can be implemented by stacking the proposed temporal convolutions with any existing EGNN layers (Thomas et al., 2018; Fuchs et al., 2020; Huang et al., 2022). These equivariant layers (EL) can be abstracted as a general class of networks that are equivariant to rotation and translation, *i.e.*,

$$\mathbf{h}^{\text{out}}, \mathbf{R} \cdot \mathbf{Z}^{\text{out}} + \boldsymbol{\mu} = \text{EL}_\theta(\mathbf{h}^{\text{in}}, \mathbf{R} \cdot \mathbf{Z}^{\text{in}} + \boldsymbol{\mu}). \tag{10}$$

In this paper, we choose the original EGNN (Satorras et al., 2021), one of the most widely adopted equivariant graph networks, as the backbone. Assuming we discretize the time window $\Delta T$ into $\{\Delta t_1, \ldots, \Delta t_P\}$ and take the current state $\mathcal{G}^{(t)}$ as input, our workflow is to first repeat its feature map by $P$ times, expand the features with time embeddings, and then forward the features into EGNO blocks composed of temporal convolutions and EGNN layers. In particular, EGNN layers would only operate on the node and channel dimension within each graph and treat the temporal dimension the same as the batch dimension. The overall framework is described in Figure 1.

A minor remaining part of the temporal convolution layer (equation (7)) is that so far it is still only equivariant to rotations but not translations. Yet, translation equivariance is desirable for some geometric features like coordinates. Indeed, this part can be trivially realized by canceling the center of mass (CoM). In detail, for temporal convolution on coordinates $\mathbf{x}$, we will first move the structure to zero-CoM by subtracting $\bar{\mathbf{x}} = \frac{1}{N} \sum_{i=1}^{N} \mathbf{x}_i$, pass it through the layer $\mathcal{T}_\theta$, and then add original CoM $\bar{\mathbf{x}}$ back. Such workflow handles the translation equivariance in a simple yet effective way.

### 4.4 TRAINING AND DECODING

**Training.** The whole framework of EGNO can be efficiently trained by minimizing the integral of the errors on decoded dynamics, *i.e.*, $\min_\theta \mathbb{E}_{\mathcal{G}^{(t)} \sim p_{\text{data}}} \int_{[0,\Delta T]} ||F_\theta(\mathcal{G}^{(t)})(\Delta t) - F^\dagger(\mathcal{G}^{(t)})(\Delta t)|| \mathrm{d}\Delta t$. In practice, we optimize the temporally discretized version:

$$\min_\theta \mathbb{E}_{\mathcal{G}^{(t)} \sim p_{\text{data}}} \frac{1}{P} \sum_{p=1}^{P} ||F_\theta(\mathcal{G}^{(t)})(\Delta t_p) - F^\dagger(\mathcal{G}^{(t)})(\Delta t_p)||, \tag{11}$$

where $\{\Delta t_1, \ldots, \Delta t_p\}$ are discrete timesteps in the time window $[0, \Delta T]$. Specifically, in this paper, we concentrate on modeling the structural dynamics, where only directional features $\mathbf{Z}$ are updated and node features $\mathbf{h}$ stay unchanged. Therefore, the norm in the objective function will only be calculated over $\mathbf{Z}$ part of the predicted structures $\mathcal{G}^{(t+\Delta t)}$.

**Decoding.** One advantage of EGNO is that all modules including temporal convolutions and equivariant networks can process data at different times in parallel. Therefore, given a current state $\mathcal{G}^{(t)}$, EGNO is capable of efficiently decoding all future structures $\mathcal{G}^{(t+\Delta t_p)}$ at timesteps $\Delta t_p \in \{\Delta t_1, \ldots, \Delta t_p\}$ with a single model call. Then a specific one can be chosen according to user preference for time scales, which in practice is a highly favorable property for studying molecular dynamics at different scales (Schreiner et al., 2023).

## 5 EXPERIMENT

In this section, we evaluate EGNO in various scenarios, including N-body simulation (§ 5.1), motion capture (§ 5.2), and molecular dynamics (§ 5.3) on small molecules (§ 5.3.1) and proteins (§ 5.3.2). In § 5.4, we perform ablation studies to investigate the necessity of our core designs in EGNO. We provide extensive visualizations in Appendix C.3.

### 5.1 N-BODY SYSTEM SIMULATIONS

**Dataset and implementation.** We adopt the 3D N-body simulation dataset (Satorras et al., 2021) which comprises multiple trajectories depicting the dynamical system formed by $N$ charged particles, with movements driven by Coulomb force. We follow the experimental setup of Satorras et al. (2021), with $N = 5$, time window $\Delta T = 10$, and 3000/2000/2000 trajectories for training/validation/testing. We use uniform discretization with $P = 5$, with more details deferred to Appendix B.1.

Table 2: F-MSE in N-body simulation *w.r.t.* different sizes of the training set.

| |Train| | 1000 | 3000 | 10000 |
|---|---|---|---|
| EGNN | 0.0094 | 0.0071 | 0.0051 |
| EGNO-U ($P = 5$) | 0.0082 | 0.0056 | 0.0036 |
| EGNO-L ($P = 5$) | 0.0071 | 0.0055 | 0.0038 |

Table 3: F-MSE in N-body simulation *w.r.t.* different numbers of time steps $P$.

| |Train|=3000 | EGNO-U | EGNO-L |
|---|---|---|
| $P = 2$ | 0.0062 | 0.0067 |
| $P = 5$ | 0.0056 | 0.0055 |
| $P = 10$ | 0.0057 | 0.0054 |

**Evaluation metrics.** We perform comparisons on two tasks, namely state-to-state (S2S) and state-to-trajectory (S2T). They measure the prediction accuracy of the final state or the whole trajectories of the decoded future dynamics within a time window, respectively. Formally, for S2S, we calculate Final Mean Squared Error (F-MSE), which is the MSE between the predicted *final* state and the ground truth, *i.e.*, F-MSE $= \|\mathbf{x}(t_P) - \mathbf{x}^\dagger(t_P)\|^2$, where $\mathbf{x}^\dagger$ is the ground truth position. For S2T, we use Average MSE (A-MSE), which instead computes the MSE averaged across all discretized time steps along the decoded trajectory, *i.e.*, A-MSE $= \frac{1}{P}\sum_{p=1}^{P} \|\mathbf{x}(t_p) - \mathbf{x}^\dagger(t_p)\|^2$. These metrics are employed throughout the experiments in this paper unless otherwise specified.

**Baselines.** For S2S, we include Linear Dynamics (Linear) (Satorras et al., 2021), SE(3)-Transformer (SE(3)-Tr.) (Fuchs et al., 2020), Tensor Field Networks (TFN) (Thomas et al., 2018), Message Passing Neural Network (MPNN) (Gilmer et al., 2017), Radial Field (RF) (Köhler et al., 2019), ClofNet (Du et al., 2022), GCPNet (Morehead & Cheng, 2023), and EGNN (Satorras et al., 2021). They ignore the temporal dependency and directly predict the last snapshot. For S2T, we extend the most competitive baseline EGNN to two variants: **EGNN-R**oll is an EGNN trained on the shortest time span and tested by iteratively rolling out (Sanchez-Gonzalez et al., 2020). **EGNN-S**equential sequentially reads out each frame of the trajectory from each EGNN layer.

**Results.** The results are listed in Table 1, where the numbers of the baselines are taken from Satorras et al. (2021). We have the following findings: **1.** Our EGNO achieves the lowest error of 0.0054

Table 1: MSE in the N-body simulation.

| | F-MSE |
|---|---|
| Linear (Satorras et al., 2021) | 0.0819 |
| SE(3)-Tr. (Fuchs et al., 2020) | 0.0244 |
| TFN (Thomas et al., 2018) | 0.0155 |
| MPNN (Gilmer et al., 2017) | 0.0107 |
| RF (Köhler et al., 2019) | 0.0104 |
| ClofNet (Du et al., 2022) | 0.0065 |
| GCPNet (Morehead & Cheng, 2023) | 0.0070 |
| EGNN (Satorras et al., 2021) | 0.0071 |
| EGNN-R | 0.0720 |
| EGNN-S | 0.0070 |
| EGNO | **0.0054** |
| | A-MSE |
| EGNN-R | 0.0215 |
| EGNN-S | 0.0045 |
| EGNO | **0.0022** |

in S2S setting, yielding an 18% relative enhancement over the most competitive baseline EGNN. EGNO also surpasses other trajectory modeling variants in S2T setting by a considerable margin. **2.** EGNN-R performs poorly compared with other trajectory modeling approaches, since the error is dramatically accumulated in each roll-out step during testing. EGNN-S is also inferior to EGNO, due to insufficient excavation of the temporal dependency along the entire dynamics trajectory.

**Data efficiency.** We compare EGNO with EGNN under different data regimes where the size of the training set sweeps over 1000, 3000, and 10000. As depicted in Table 2, EGNOs achieve much lower simulation error than EGNN in all scenarios, and are observed to be nearly 3× more data-efficient.

**The approach of temporal discretization.** We study two methods of discretization for the trajectory, namely EGNO-U, which *uniformly* samples $\{t_p\}_{p=1}^{P}$ with $t_p = t + \frac{p}{P}\Delta T$, and EGNO-L, which selects the *last* $P$ snapshots at the tail of the trajectory with interval $\delta$, *i.e.*, $t_p = t + \Delta T - \delta(P - p)$. In Table 2 and 3, We find that EGNO obtains promising performance regardless of the discretization method. EGNO-L performs better than EGNO-U in terms of F-MSE when data is scarce since it gives a more detailed description of the part near the end of the trajectory. However, EGNO-U becomes better when $P = 2$, because uniform discretization is more informative in depicting the entire dynamics. We also provide results and analyses on other datasets in Appendix C.2.

**The number of discretized time steps.** To investigate how the granularity of the temporal discretization influences the performance, we switch the number of time steps $P$ in training within $\{2, 5, 10\}$. Interestingly, in Table 3, EGNO with $P = 2$ outperforms EGNN, while the performance is further boosted when $P$ is increased to 5, which aligns with our proposal of leveraging trajectory modeling over snapshots. The MSEs of $P = 10$ remain close to $P = 5$ with no clear enhancement, possibly

Table 5: MSE ($\times 10^{-2}$) on MD17 dataset. Upper part: F-MSE for `S2S`. Lower part: A-MSE for `S2T`.

| | Aspirin | Benzene | Ethanol | Malonaldehyde | Naphthalene | Salicylic | Toluene | Uracil |
|---|---|---|---|---|---|---|---|---|
| RF | $10.94_{\pm 0.01}$ | $103.72_{\pm 1.29}$ | $4.64_{\pm 0.01}$ | $13.93_{\pm 0.03}$ | $0.50_{\pm 0.01}$ | $1.23_{\pm 0.01}$ | $10.93_{\pm 0.04}$ | $0.64_{\pm 0.01}$ |
| TFN | $12.37_{\pm 0.18}$ | $58.48_{\pm 1.98}$ | $4.81_{\pm 0.04}$ | $13.62_{\pm 0.08}$ | $0.49_{\pm 0.01}$ | $1.03_{\pm 0.02}$ | $10.89_{\pm 0.01}$ | $0.84_{\pm 0.02}$ |
| SE(3)-Tr. | $11.12_{\pm 0.06}$ | $68.11_{\pm 0.67}$ | $4.74_{\pm 0.13}$ | $13.89_{\pm 0.02}$ | $0.52_{\pm 0.01}$ | $1.13_{\pm 0.02}$ | $10.88_{\pm 0.06}$ | $0.79_{\pm 0.02}$ |
| EGNN | $14.41_{\pm 0.15}$ | $62.40_{\pm 0.53}$ | $4.64_{\pm 0.01}$ | $13.64_{\pm 0.01}$ | $0.47_{\pm 0.02}$ | $1.02_{\pm 0.02}$ | $11.78_{\pm 0.07}$ | $0.64_{\pm 0.01}$ |
| EGNN-R | $14.51_{\pm 0.19}$ | $62.61_{\pm 0.75}$ | $4.94_{\pm 0.21}$ | $17.25_{\pm 0.05}$ | $0.82_{\pm 0.02}$ | $1.35_{\pm 0.02}$ | $11.59_{\pm 0.04}$ | $1.11_{\pm 0.02}$ |
| EGNN-S | $9.50_{\pm 0.10}$ | $66.45_{\pm 0.89}$ | $4.63_{\pm 0.01}$ | $12.88_{\pm 0.01}$ | $0.45_{\pm 0.01}$ | $1.00_{\pm 0.02}$ | $10.78_{\pm 0.05}$ | $0.60_{\pm 0.01}$ |
| EGNO | $\mathbf{9.18}_{\pm 0.06}$ | $\mathbf{48.85}_{\pm 0.55}$ | $\mathbf{4.62}_{\pm 0.01}$ | $\mathbf{12.80}_{\pm 0.02}$ | $\mathbf{0.37}_{\pm 0.01}$ | $\mathbf{0.86}_{\pm 0.02}$ | $\mathbf{10.21}_{\pm 0.05}$ | $\mathbf{0.52}_{\pm 0.02}$ |
| EGNN-R | $12.07_{\pm 0.11}$ | $23.73_{\pm 0.30}$ | $3.44_{\pm 0.17}$ | $13.38_{\pm 0.03}$ | $0.63_{\pm 0.01}$ | $1.15_{\pm 0.02}$ | $5.04_{\pm 0.02}$ | $0.89_{\pm 0.01}$ |
| EGNN-S | $9.49_{\pm 0.12}$ | $29.99_{\pm 0.65}$ | $3.29_{\pm 0.01}$ | $11.21_{\pm 0.01}$ | $0.43_{\pm 0.01}$ | $1.36_{\pm 0.02}$ | $4.85_{\pm 0.04}$ | $0.68_{\pm 0.01}$ |
| EGNO | $\mathbf{7.37}_{\pm 0.07}$ | $\mathbf{22.41}_{\pm 0.31}$ | $\mathbf{3.28}_{\pm 0.02}$ | $\mathbf{10.67}_{\pm 0.01}$ | $\mathbf{0.32}_{\pm 0.01}$ | $\mathbf{0.77}_{\pm 0.01}$ | $\mathbf{4.58}_{\pm 0.03}$ | $\mathbf{0.47}_{\pm 0.01}$ |

because the trajectory with $P = 5$ has already been sufficiently informative for its geometric pattern to be abundantly captured by the equivariant Fourier temporal convolution modules.

## 5.2 MOTION CAPTURE

**Dataset and implementation.** We further benchmark our model on CMU Motion Capture dataset (CMU, 2003), which involves 3D trajectories captured from various human motion movements. We focus on two motions: Subject #35 (`Walk`) and Subject #9 (`Run`), following the setups and data splits in Huang et al. (2022); Han et al. (2022). Similar to N-body simulation, the input includes initial positions and velocities of the intersections, and $\Delta T = 30$. We use $P = 5$ and uniform discretization by default. More details are in Appendix B.

**Results.** As exhibited in Table 4, our EGNO surpasses the baselines by a large margin on both `Walk` and `Run` datasets and in both `S2S` and `S2T` cases, *e.g.*, up to around $52\%$ on average in `S2S`. The error propagation of EGNN-R becomes more severe since the motion capture involves more complex and varying dynamics than N-body simulation. EGNN-S generally performs comparably to other `S2S` methods, while still being inferior to EGNO which models the entire trajectory compactly.

Table 4: MSE ($\times 10^{-2}$) on the motion capture dataset. The upper part is F-MSE for `S2S` and the lower part is A-MSE for `S2T`.

| | Subject #35 Walk | Subject #9 Run |
|---|---|---|
| MPNN | $36.1 \pm 1.5$ | $66.4 \pm 2.2$ |
| RF | $188.0 \pm 1.9$ | $521.3 \pm 2.3$ |
| TFN | $32.0 \pm 1.8$ | $56.6 \pm 1.7$ |
| SE(3)-Tr. | $31.5 \pm 2.1$ | $61.2 \pm 2.3$ |
| EGNN | $28.7 \pm 1.6$ | $50.9 \pm 0.9$ |
| EGNN-R | $90.7 \pm 2.4$ | $816.7 \pm 2.7$ |
| EGNN-S | $26.4 \pm 1.5$ | $54.2 \pm 1.9$ |
| EGNO | $\mathbf{8.1} \pm 1.6$ | $\mathbf{33.9} \pm 1.7$ |
| EGNN-R | $32.0 \pm 1.6$ | $277.3 \pm 1.8$ |
| EGNN-S | $14.3 \pm 1.2$ | $28.5 \pm 1.3$ |
| EGNO | $\mathbf{3.5} \pm 0.5$ | $\mathbf{14.9} \pm 0.9$ |

## 5.3 MOLECULAR DYNAMICS

### 5.3.1 SMALL MOLECULES

**Dataset.** We adopt MD17 (Chmiela et al., 2017) dataset to evaluate the capability of our EGNO on modeling molecular dynamics. The dataset consists of the molecular dynamics trajectories of eight small molecules, including aspirin, benzene, *etc*. We follow the setup and split by Huang et al. (2022) which randomly partitions each trajectory into 500/2000/2000 subsets for training/validation/testing while $\Delta T$ is chosen to be 3000. We use $P = 8$ and uniform discretization by default.

**Results.** The results are illustrated in Table 5. EGNO obtains the best performance on all eight molecules, verifying the applicability of EGNO towards modeling molecular dynamics. The performance gain is most notable on Aspirin, one of the most complicated structures on MD17, where EGNO, with an MSE of $9.18 \times 10^{-2}$, outperforms its backbone EGNN by $36\%$. Compared with EGNN-R/S, EGNO consistently gives more accurate predictions for both `S2S` and `S2T` evaluations, showcasing the importance of conducting equivariant temporal convolution in geometric space.

### 5.3.2 PROTEINS

**Dataset and implementation.** We use the Adk equilibrium trajectory dataset (Seyler & Beckstein, 2017) integrated in the MDAnalysis (Richard J. Gowers et al., 2016) toolkit, which depicts the

molecular dynamics trajectory of apo adenylate kinase. We follow the setting and split of Han et al. (2022). We use $P = 4$ by default. We additionally consider EGHN (Han et al., 2022), the state-of-the-art model on this benchmark, as a baseline. To investigate the compatibility of our approach with different backbones, we incorporate equivariant temporal convolution into EGHN by inserting it to the end of each block in EGHN (details in Appendix B.4) and name it EGHNO.

**Results.** The results are presented in Table 6. Equipped with our equivariant temporal convolution, EGHNO achieves state-of-the-art on this challenging benchmark. Furthermore,

Table 6: F-MSE on AdK equilibrium trajectory dataset.

| Linear | RF | MPNN | EGNN | EGHN | EGNO | EGHNO |
|--------|------|-------|-------|-------|-------|--------|
| 2.890 | 2.846 | 2.322 | 2.735 | 2.034 | 2.231 | **1.801** |

both EGNO and EGHNO offer significant increment to the performance to their original backbones, *i.e.*, EGNN and EGHN, showing that our equivariant temporal convolution is highly compatible with various backbones, indicating its broad applicability to a diverse range of models and tasks.

## 5.4 ABLATION STUDIES

We conduct ablation studies on the N-body system simulation task and motion capture dataset to inspect the role of the designed parts, with the results summarized in Table 7.

**Incorporating geometric information in temporal convolution.** In § 4.2, we propose to involve temporal convolution on geometric features $[\mathbf{h}, \mathbf{Z}]$ in frequency domain. To investigate its importance, we implement three variants of EGNO: variant IV discards the entire temporal convolution and reduces to an EGNN with time embeddings; variant III only processes the invariant feature $\mathbf{h}$ in the temporal convolution; variant II considers $\mathbf{h}, \mathbf{x}$ but neglects $\mathbf{v}$. Interestingly, compared with the complete version of EGNO (labeled by I) which incorporates $\mathbf{h}, \mathbf{x}, \mathbf{v}$, variant II discards the velocity $\mathbf{v}$ in temporal convolution, which brings performance detriment on both datasets. By further removing all directional informa-

Table 7: Ablation studies on N-body simulation and Mocap-Run datasets. Numbers refer to the F-MSE ($\times 10^{-2}$ for Run). Columns with $\mathbf{h}, \mathbf{x}, \mathbf{v}$ indicate whether such geometric information is processed in equation 7.

| | | $\mathbf{h}$ | $\mathbf{x}$ | $\mathbf{v}$ | N-body | Mocap-Run |
|------|-----|-----|-----|-----|--------|-----------|
| EGNO | I | ✓ | ✓ | ✓ | **0.0055** | **33.9** |
| | II | ✓ | ✓ | - | 0.0057 | 35.6 |
| | III | ✓ | - | - | 0.0061 | 39.1 |
| | IV | - | - | - | 0.0072 | 48.2 |
| EGNN | | | | | 0.0071 | 50.9 |

tion, variant III, with invariant $\mathbf{h}$ only, incurs much worse performance. Without temporal convolution (IV), the model cannot sufficiently capture temporal patterns and suffers from the worst performance. Yet and still, all variants outperform EGNN remarkably thanks to geometric temporal convolution.

**Number of modes.** We study the influence of the number of modes $I$ reserved in equivariant Fourier convolution with results in Figure 2. The simulation errors are decreased notably when $I$ is increased from 0 to 2, with more temporal patterns with geometric information captured and encoded in the convolution. When $I$ is further increased to 3, the performance improves marginally on Mocap-Run or even becomes worse on N-body, potentially because redundant frequencies, which encode noisy patterns, are encapsulated and thus lead to overfitting.

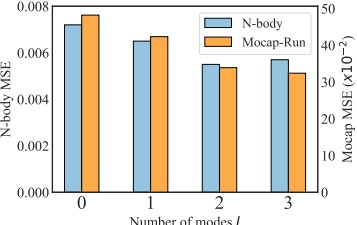

Figure 2: Ablations on $I$.

## 6 CONCLUSION

In this paper, we present equivariant graph neural operator (EGNO), a principled method that models physical dynamics by explicitly considering the temporal correlations within it. Our key innovation is to formulate the dynamics as an equivariant function describing state evolution over time and learn neural operators to approximate it. To this end, we develop a novel equivariant temporal convolution layer parameterized in the Fourier space. Comprehensive experiments demonstrate its significantly superior performance in modeling geometric dynamics. With EGNO as a general framework, future work includes extending EGNO to other physical dynamics domains such as astronomical objects, or scaling up EGNO to more challenging dynamics such as fluids and deformable materials.

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

## A  FORMAL STATEMENTS AND PROOFS

### A.1  STATEMENT OF UNIVERSALITY

We include the formal statement from Kovachki et al. (2021a;b) for the modeling capacity of EGNO here to make the paper self-contained.

**Proposition A.1.** *The neural operators defined in equation* (1) *approximates the solution operator of the physical dynamics ODE equation* (4), *i.e., the mapping from a current structure $\mathcal{G}^{(t)}$ to future trajectory $\{\mathcal{G}^{(t+\Delta t)} : \Delta t \sim [0, \Delta T]\}$, arbitrarily well.*

### A.2  PROOF OF THEOREM 4.1

In this section, we prove Theorem 4.1, which justifies that our proposed architecture is SO(3)-equivariant. We recall that our action is defined on functions $f = [f_{\mathbf{h}}, f_{\mathbf{Z}}]^{\mathrm{T}} : D \to \mathbb{R}^{N \times (k+m \times 3)}$ (where $f_{\mathbf{h}}$ is the $D \to \mathbb{R}^{N \times k}$ node feature component and $f_{\mathbf{Z}} : D \to \mathbb{R}^{N \times m \times 3}$ is the spatial feature component) by

$$(\mathbf{R}f)(t) = [f_{\mathbf{h}}(t), \mathbf{R}f_{\mathbf{Z}}(t)]^{\mathrm{T}}. \tag{12}$$

We first analyze the action of the Fourier transform, showing that it is equivariant:

**Lemma A.2** (Fourier Action Equivariance). *The actions of the Fourier and Inverse Fourier Transform are SO(3)-equivariant.*

*Proof.* $\mathcal{F}$ is a dimension-wise Fourier transform that maps from $(\mathbb{R}^D \to \mathbb{R}^{\mathcal{G}}) \to \mathbb{C}^{I \times \mathcal{G}}$, where $I$ is the truncation on the number of the Fourier coefficients. For our purposes, this maps our input function $f : D \to \mathbb{R}^{N \times (k+m \times 3)}$ to $\mathbb{C}^{I \times (k+m \times 3)}$ space[4]. The action of $\mathbf{R}$ is the standard matrix-tensor action on the $\mathcal{F}f_{\mathbf{Z}} \in \mathbb{C}^{I \times m \times 3}$ component

$$\mathbf{R} \cdot \mathcal{F}f = \mathbf{R} \cdot [\mathcal{F}f_{\mathbf{h}}, \mathcal{F}f_{\mathbf{Z}}]^{\mathrm{T}} = [\mathcal{F}f_{\mathbf{h}}, \mathbf{R} \cdot \mathcal{F}f_{\mathbf{Z}}]^{\mathrm{T}} \tag{13}$$

Conversely, we have that

$$\mathcal{F}(\mathbf{R} \cdot f) = \mathcal{F}[f_{\mathbf{h}}, \mathbf{R} \cdot f_{\mathbf{Z}}]^{\mathrm{T}} = [\mathcal{F}f_{\mathbf{h}}, \mathcal{F}(\mathbf{R} \cdot f_{\mathbf{Z}})]^{\mathrm{T}} \tag{14}$$

We need to show that $\mathbf{R} \cdot \mathcal{F}f_{\mathbf{Z}} = \mathcal{F}(\mathbf{R} \cdot f_{\mathbf{Z}})$. This can be done by recalling that the Fourier transform $\mathcal{F}$ is a linear operator (even when it is discrete as given here). As such, we can write out the equivariance directly here on a dimension-wise basis:

$$(\mathbf{R} \cdot \mathcal{F}f_{\mathbf{Z}})_{ij \cdot} = \mathbf{R} \begin{bmatrix} (\mathcal{F}f_{\mathbf{Z}})_{ij1} \\ (\mathcal{F}f_{\mathbf{Z}})_{ij2} \\ (\mathcal{F}f_{\mathbf{Z}})_{ij3} \end{bmatrix} \tag{15}$$

$$= \begin{bmatrix} R_{11}(\mathcal{F}f_{\mathbf{Z}})_{ij1} + R_{12}(\mathcal{F}f_{\mathbf{Z}})_{ij2} + R_{13}(\mathcal{F}f_{\mathbf{Z}})_{ij3} \\ R_{21}(\mathcal{F}f_{\mathbf{Z}})_{ij1} + R_{22}(\mathcal{F}f_{\mathbf{Z}})_{ij2} + R_{23}(\mathcal{F}f_{\mathbf{Z}})_{ij3} \\ R_{31}(\mathcal{F}f_{\mathbf{Z}})_{ij1} + R_{32}(\mathcal{F}f_{\mathbf{Z}})_{ij2} + R_{33}(\mathcal{F}f_{\mathbf{Z}})_{ij3} \end{bmatrix} \tag{16}$$

$$= \begin{bmatrix} \mathcal{F}((R_{11}f_{\mathbf{Z}})_{ij1} + (R_{12}f_{\mathbf{Z}})_{ij2} + (R_{13}f_{\mathbf{Z}})_{ij3}) \\ \mathcal{F}((R_{21}f_{\mathbf{Z}})_{ij1} + (R_{22}f_{\mathbf{Z}})_{ij2} + (R_{23}f_{\mathbf{Z}})_{ij3}) \\ \mathcal{F}((R_{31}f_{\mathbf{Z}})_{ij1} + (R_{32}f_{\mathbf{Z}})_{ij2} + (R_{33}f_{\mathbf{Z}})_{ij3}) \end{bmatrix} \tag{17}$$

$$= (\mathcal{F} \cdot (\mathbf{R}f_{\mathbf{Z}}))_{ij \cdot}. \tag{18}$$

which is the desired $\mathcal{F}$ equivariance.

To show that the inverse Fourier transform $\mathcal{F}^{-1}$ acts equivariantly, we could easily apply the same linearity condition on $\mathcal{F}^{-1}$ and go through the details. However, we can do this more directly through a categorical argument:

$$\mathcal{F}(\mathbf{R} \cdot \mathcal{F}^{-1}f_{\mathbf{Z}}) = \mathbf{R} \cdot \mathcal{F}\mathcal{F}^{-1}f_{\mathbf{Z}} = \mathbf{R} \cdot f_{\mathbf{Z}} \implies \mathbf{R} \cdot \mathcal{F}^{-1}f_{\mathbf{Z}} = \mathcal{F}^{-1}(\mathbf{R} \cdot f_{\mathbf{Z}})$$

Note that $\mathcal{F}^{-1}$ is only right inverse of $\mathcal{F}$ and is only a full inverse when the function domain is restricted to all functions with trivial Fourier coefficients for degrees $> I$. However, it can be applied on the left side here since $\mathbf{R} \cdot \mathcal{F}^{-1}f_{\mathbf{Z}}$ is in this function class since this class is closed under scalar multiplication and addition (so it is closed under the action of $\mathbf{R}$). $\qquad\square$

---

[4]Note that, the Fourier transform only operates on the temporal dimension and the node dimension $N$ can be trivially regarded as the same as the batch dimension, so we omit that dimension in our proof.

This leads directly to the full proof of the theorem.

*Proof of Theorem 4.1.* Recall that our neural operator is given in full form by

$$\mathcal{T}_\theta f = f + \sigma \circ (\mathcal{F}^{-1}(\begin{bmatrix} \mathbf{M}_\theta^{\mathbf{h}} & \mathbf{0} \\ \mathbf{0} & \mathbf{M}_\theta^{\mathbf{Z}} \end{bmatrix} \cdot (\mathcal{F} \begin{bmatrix} f_{\mathbf{h}} \\ f_{\mathbf{Z}} \end{bmatrix}))) \tag{19}$$

The equivariance is shown directly

$$\mathbf{R} \cdot \mathcal{T}_\theta f = \mathbf{R} \cdot \left( f + \sigma \circ (\mathcal{F}^{-1}(\begin{bmatrix} \mathbf{M}_\theta^{\mathbf{h}} & \mathbf{0} \\ \mathbf{0} & \mathbf{M}_\theta^{\mathbf{Z}} \end{bmatrix} \cdot (\mathcal{F} \begin{bmatrix} f_{\mathbf{h}} \\ f_{\mathbf{Z}} \end{bmatrix}))) \right) \tag{20}$$

$$= \mathbf{R} \cdot f + \mathbf{R} \cdot \sigma \circ (\mathcal{F}^{-1}(\begin{bmatrix} \mathbf{M}_\theta^{\mathbf{h}} & \mathbf{0} \\ \mathbf{0} & \mathbf{M}_\theta^{\mathbf{Z}} \end{bmatrix} \cdot (\mathcal{F} \begin{bmatrix} f_{\mathbf{h}} \\ f_{\mathbf{Z}} \end{bmatrix}))) \tag{21}$$

$$= \mathbf{R} \cdot f + \sigma \circ \mathbf{R} \cdot (\mathcal{F}^{-1}(\begin{bmatrix} \mathbf{M}_\theta^{\mathbf{h}} & \mathbf{0} \\ \mathbf{0} & \mathbf{M}_\theta^{\mathbf{Z}} \end{bmatrix} \cdot (\mathcal{F} \begin{bmatrix} f_{\mathbf{h}} \\ f_{\mathbf{Z}} \end{bmatrix}))) \tag{22}$$

$$= \mathbf{R} \cdot f + \sigma \circ (\mathcal{F}^{-1}(\mathbf{R} \cdot \begin{bmatrix} \mathbf{M}_\theta^{\mathbf{h}} & \mathbf{0} \\ \mathbf{0} & \mathbf{M}_\theta^{\mathbf{Z}} \end{bmatrix} \cdot (\mathcal{F} \begin{bmatrix} f_{\mathbf{h}} \\ f_{\mathbf{Z}} \end{bmatrix}))) \tag{23}$$

$$= \mathbf{R} \cdot f + \sigma \circ (\mathcal{F}^{-1}(\begin{bmatrix} \mathbf{M}_\theta^{\mathbf{h}} & \mathbf{0} \\ \mathbf{0} & \mathbf{M}_\theta^{\mathbf{Z}} \end{bmatrix} \cdot \mathbf{R} \cdot (\mathcal{F} \begin{bmatrix} f_{\mathbf{h}} \\ f_{\mathbf{Z}} \end{bmatrix}))) \tag{24}$$

$$= \mathbf{R} \cdot f + \sigma \circ (\mathcal{F}^{-1}(\begin{bmatrix} \mathbf{M}_\theta^{\mathbf{h}} & \mathbf{0} \\ \mathbf{0} & \mathbf{M}_\theta^{\mathbf{Z}} \end{bmatrix} \cdot (\mathcal{F} \mathbf{R} \cdot \begin{bmatrix} f_{\mathbf{h}} \\ f_{\mathbf{Z}} \end{bmatrix}))) \tag{25}$$

$$= \mathcal{T}_\theta(\mathbf{R} \cdot f) \tag{26}$$

$$\square$$

Line 22 follows since $\sigma$ is an identity operator on $\mathbf{Z}$ component. Lines 23 and 25 are the inverse and regular Fourier transforms and follows from Lemma A.2. The only new line is 24, which follows since $\mathbf{M}_\theta^{\mathbf{Z}}$ is a scalar multiplication for each component (which commutes with the matrix multiplication from $\mathbf{R}$). Therefore, we have that our neural operator is equivariant, as desired.

## B EXPERIMENT DETAILS

### B.1 DATASET DETAILS

**N-body Simulation.** Originally introduced in Kipf et al. (2018) and further extended to the 3D version by Satorras et al. (2021), the N-body simulation comprises multiple trajectories, each of which depicts a dynamical system formed by $N$ charged particles with their movements driven by the interacting Coulomb force. For each trajectory, the inputs are the charges, initial positions, and velocities of the particles. We follow the experimental setup of Satorras et al. (2021), with $N = 5$, time window $\Delta T = 10$, and 3000 trajectories for training, 2000 for validation, and 2000 for testing. We take $P = 5$ by default. In accordance with Satorras et al. (2021), the input node feature is instantiated as the magnitude of the velocity $\|\mathbf{v}\|_2$, the edge feature is specified as $c_i c_j$ where $c_i, c_j$ are the charges, and the graph is constructed in a fully-connected manner without self-loops.

**MD17.** MD17 (Chmiela et al., 2017) data consists of the molecular dynamics trajectories of eight small molecules. We randomly split each one into train/validation/test sets with 500/2000/2000 pairs of state and later trajectories, respectively. We choose $\Delta T = 5000$ as the time window between the input state and the last snapshot of the prediction trajectories, and calculate the difference between each as the input velocity. We also compute the norm of velocities and concatenate them with the atom type as the node feature. We follow the conventions in this field to remove the hydrogen atoms and focus on the dynamics of heavy atoms. For the graph structure, we follow previous studies (Shi et al., 2021; Xu et al., 2022) to expand the original molecular graph by connecting 2-hop neighbors. Then we take the concatenation of the hop type, the atomic types of connected nodes, and the chemical bond type as the edge feature.

Table 8: Summary of hyperparameters for EGNO on all datasets.

|          | batch | lr   | wd    | layer | hidden | timestep | time_emb | num_mode |
|----------|-------|------|-------|-------|--------|----------|----------|----------|
| N-body   | 100   | 1e-4 | 1e-8  | 4     | 64     | 5        | 32       | 2        |
| Walk/Run | 12    | 5e-4 | 1e-10 | 6     | 128    | 5        | 32       | 2        |
| MD17     | 100   | 1e-4 | 1e-15 | 5     | 64     | 8        | 32       | 2        |
| Protein  | 8     | 5e-5 | 1e-4  | 4     | 128    | 4        | 32       | 2        |

**CMU Motion Capture.** CMU Motion Capture dataset (CMU, 2003) involves 3D trajectories captured from various human motion movements. We focus on two motions: Subject #35 (Walk) and Subject #9 (Run), following the setups and data splits in Huang et al. (2022); Han et al. (2022). Subject #35 contains 200/600/600 trajectories for training/validation/testing, while Subject #9 contains 200/240/240. We view the joints as edges and their intersections (31 in total) as nodes. Similar to N-body simulation, the input includes initial positions and velocities of the intersections, and $\Delta T = 30$. We use $P = 5$ and uniform discretization by default.

**Protein.** We use the preprocessed version (Han et al., 2022) of the Adk equilibrium trajectory dataset (Seyler & Beckstein, 2017) integrated in the MDAnalysis (Richard J. Gowers et al., 2016) toolkit. In detail, the AdK equilibrium dataset depicts the molecular dynamics trajectory of apo adenylate kinase with CHARMM27 force field (MacKerell Jr et al., 2000), simulated with explicit water and ions in NPT at 300 K and 1 bar. The meta-data is saved every 240 ps for a total of 1.004 $\mu$s. We adopt the split by Han et al. (2022) which divides the entire trajectory into a training set with 2481 sub-trajectories, a validation set with 827, and a testing set with 878 trajectories, respectively. The backbone of the protein is extracted, and the graph is constructed using cutoff with radius 10Å.

### B.2 MORE IMPLEMENTATION DETAILS

**Time embedding.** In EGNO, we need to make the model aware of the time position for the structures in the trajectory. To this end, we add "time embeddings" to the input features. We implement the time embedding with sine and cosine functions of different frequencies, following the sinusoidal positional encoding in Transformer (Vaswani et al., 2017). For timestep $\Delta t_i$, the time embedding is implemented by:

$$
\begin{aligned}
emb_{2j} &= sin(i/10000^{2j/d_{\text{emb}}}), \\
emb_{2j+1} &= cos(i/10000^{2j/d_{\text{emb}}}),
\end{aligned}
\tag{27}
$$

where $d_{\text{emb}}$ denotes the dimension defined for time embeddings, as shown in Table 8.

**Fast Fourier Transform.** In this paper, the FFT algorithm is realized by PyTorch implementation. While FFT algorithms are often more efficient when applied to sequences whose lengths are powers of 2, many FFT implementations, including PyTorch implementation, have optimizations that allow them to efficiently process time series of arbitrary lengths.

The requirement for power-of-2 lengths is related to certain specific FFT algorithms rather than a fundamental limitation of the FFT itself. In many cases like PyTorch, the FFT implementation is designed with the Cooley-Tukey FFT Algorithm for General Factorizations, which can handle sequences of arbitrary length, including those that are not powers of 2.

**Complex numbers.** In our paper, the Fourier kernel in equations (7) and (8) are defined as complex tensors. We implement the complex tensor with two tensors to represent the real and imaginary components of complex numbers, and use torch.view_as_complex to convert the two tensors into the complex one. The computation is roughly as efficient as typical real tensors.

### B.3 HYPERPARAMETERS

We provide detailed hyperparameters of our EGNO in Table 8. Specifically, batch is for batch size, lr for learning rate, wd for weight decay, layer for the number of layers, hidden for hidden dimension, timestep for the number of time steps, time_emb for the dimension of time embedding, num_mode for the number of modes (frequencies). We adopt Adam optimizer (Kingma & Ba, 2014) and all models are trained towards convergence with an earlystopping of 50 epochs on

the validation loss. In particular, for EGHNO, we require additional hyperparemeters: the number of pooling layers is 4, the number of decoding layers is 2, and the number of message passing layer is 4, the same as Han et al. (2022). For the baselines, we strictly follow the setup in previous works and report the numbers from them in the S2S setting. For S2T, we keep the backbone EGNN the same number of layers and hidden dimension as ours for EGNN-R and EGNN-S, ensuring fairness.

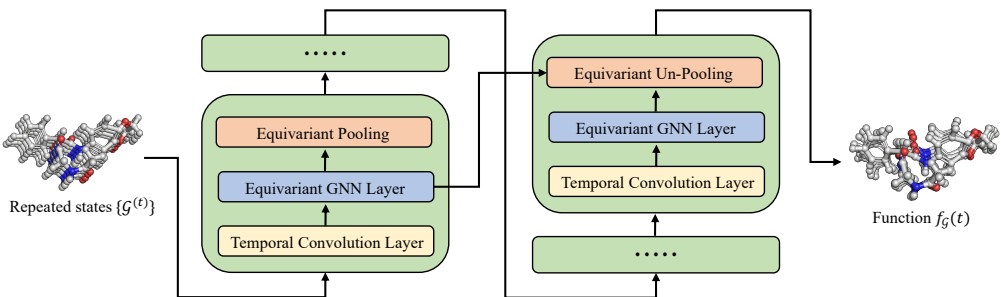

Figure 3: Illustration of EGHNO. Here, we omit the input state repetition and time embedding conditioning process in Figure 1 and concentrate on the model details.

### B.4 Model Details of EGHNO

Here we provide an illustration of EGHNO in Figure 3 to show the details of its architecture. In addition to the equivariant GNN layers, we have the equivariant pooling and unpooling layers introduced in the equivariant graph hierarchical network (EGHN) (Han et al., 2022). To realize EGHNO, we also stack the temporal convolution layers before each EGNN layer similar to EGNO (Figure 1). In particular, we follow EGHN and add jump connections between the equivariant unpooling layer and the corresponding low-level pooling layer, which helps to capture the low-level details of large geometric systems.

## C More Experiment Results

### C.1 More ablation studies

We provide more ablation studies in this section to justify the capacity of EGNO.

**Trajectory-to-trajectory.** Instead of inputting repetitions of $G(t)$, we can also input a sequence of previous structures to the model, which leads to a trajectory-to-trajectory (T2T) model. We test this model on N-body simulation and report the result as follows, named EGNO-T2T. The A-MSE on the predicted trajectory is reported in table 9.

Table 9: Average MSE in the N-body simulation. Results of EGNN-R, EGNN-S, and EGNO are directly taken from Table 1 and the experiments share the same setup.

|       | EGNN-R | EGNN-S | EGNO  | EGNO-T2T | EGNO-TB |
|-------|--------|--------|-------|----------|---------|
| A-MSE | 0.0215 | 0.0045 | 0.0022 | 0.0020   | 0.0039  |

As shown in the result, our EGNO can handle the T2T task and even achieve better results. However, we emphasize that, although this implementation achieved better results than the original EGNO, we do not want to highlight this because EGNO and all baseline models only operate on a single input point and EGNO-T2T makes use of more input information. Therefore, we take this just as an ablation study and leave detailed investigations of the T2T framework as a promising future direction.

**Temporal Bundling.** Brandstetter et al. (2022) introduces the temporal bundling (TB) method, which tackles the state-to-trajectory task in an orthogonal direction, by factorizing future trajectory. We view TB as a mixture of our designed EGNN-R baseline and EGNO-IV variant in ablation study (table 7): EGNN-R learns to predict trajectory by rolling out and EGNO-IV learns to predict trajectory in parallel with a single model call, while TB proposed to factorize trajectory as multiple blocks and learns to predict blocks one by one. We tested such EGNN-TB on the N-body simulation benchmark by factorizing trajectory into blocks of size K=2. The A-MSE result is reported in table 9, which is the accumulated error for unrolling trajectory.

Table 11: MSE ($\times 10^{-2}$) on MD17 dataset.

| | Aspirin | Benzene | Ethanol | Malonaldehyde | Naphthalene | Salicylic | Toluene | Uracil |
|---|---|---|---|---|---|---|---|---|
| EGNO-U | 9.18 | 48.85 | 4.62 | 12.80 | 0.37 | 0.86 | 10.21 | 0.52 |
| EGNO-L | 9.32 | 57.39 | 4.62 | 12.80 | 0.35 | 0.84 | 10.25 | 0.57 |

As shown in the result, EGNN-TB is indeed more expressive than EGNN-R and EGNN-S. However, our method still achieved much better results, which demonstrates the effectiveness of the proposed Fourier method to directly model the temporal correlation over the whole sequence.

However, we want to highlight two points here: 1) Brandstetter et al. (2022) is related to EGNO on the temporal modeling side. Another key contribution of EGNO is imposing geometric symmetry into the model, making EGNO fundamentally different from Brandstetter et al. (2022). 2) Even for temporal modeling, the TB method actually is orthogonal to the Fourier-based method, where we can still factorize predictions as blocks while using Fourier operators to predict each block. Therefore, these two methods are not in conflict and we leave the novel combination as future works.

**Neural ODE.** For modeling dynamics as a trajectory, another approach is to learn neural ODEs as surrogates to modeling the gradient field between two states, which capture the temporal correlation by reparameterizing an ODE. ODE and Fourier operator learning can be viewed as two orthogonal methods to model the dynamics. In this section, we first present empirical comparisons, and then discuss the methodological differences and connections.

We follow the setup described in Appendix A.7.2 of Du et al. (2022) to implement an EGNN trained with ODE objectives, named EGNN-ODE, and test the methods by their interpolation (inter) and extrapolation (extra) settings in our N-body simulation task. For EGNO, we evaluate the interpolation by sampling more timesteps to query the trajectory from the output function, implemented by repeating the input more times and concatenating extra time embeddings specified by the additionally queried intermediate timesteps, similar to our approach in the experiment of Appendix C.4. The extrapolation is evaluated by unrolling the prediction of the dynamics towards the additionally queried timesteps. In this way, both the ODE formulation and our Fourier operator learning approach are capable of handling interpolation and extrapolation scenarios. We report the A-MSE results for both inter and extra trajectories in Table 10.

Empirically, as shown in the results, the errors of using ODE for both inter and extra are higher than EGNO, due to the error accumulated during solving step by step. By contrast, EGNO yields the prediction over a sequence of structures within the trajectory in parallel with one model call, which mitigates error accumulation in the iterative ODE-solving procedure. Moreover, when conducting the experiment we observed that leveraging ODE indeed introduced significant computation overhead in both training and inference especially when the step size is

Table 10: Average MSE in the N-body simulation. **Inter** and **Extra** denote the interpolation and extrapolation task respectively.

| Method | Inter | Extra |
|---|---|---|
| EGNN-ODE | 0.0034 | 0.0620 |
| EGNO | 0.0022 | 0.0358 |

small, while our EGNO, by modeling the trajectory in the Fourier domain, enables parallel decoding and thus enjoys very high efficiency for both training and inference.

Methodologically, indeed, we realize there is even potential to combine ODE and FNO. To be more specific, we can also roll out the dynamics with EGNO for additional timesteps, and utilize the closest predicted point as the initial value to solve the ODE towards other test-time queried time steps. This could potentially offer better prediction accuracy as well as efficiency for ODEs. We leave further studies of the novel combination as future work.

## C.2 DISCRETIZATION METHODS

We present more results and discussions on the selection of discretization methods. On Motion Capture, EGNO-U achieves F-MSE of 8.1 on `Walk` and 33.9 on `Run`, while EGNO-L obtains 11.2 on `Walk` and 31.6 on `Run`. The results on MD17 are depicted in Table 11. On protein data, EGNO-U has an F-MSE of 2.231, EGNO-L has an F-MSE of 2.231, EGHNO-U has an F-MSE of 1.801, and EGHNO-L has an F-MSE of 1.938. On Motion Capture, EGNO-U performs better than EGNO-L on `Walk`, while it becomes slightly worse than EGNO-L for the other, *i.e.*, `Run`. We speculate that

since running incurs more significant vibrations of the positions in the movements, EGNO-L, by focusing more on the last short period of the trajectory, permits better generalization than EGNO-U, which instead operates on a uniform sampling along the entire trajectory, whereas on the more stable dataset, walking, the prevalence of EGNO-U is observed. EGNO-U and EGNO-L yield similar simulation error on MD17 and protein data, since the molecular trajectories are relatively stable with small vibrations around the metastable state, making the temporal sampling less sensitive.

### C.3 QUALITATIVE VISUALIZATIONS

**N-body and Motion Capture.** We provide visualizations of the dynamics predicted by EGNO in this section. The results of particle simulations, Mocap-`Walk`, and Mocap-`Run` are shown in Figures 4 to 6 respectively. As shown in these figures, our EGNO can produce not only accurate final snapshot predictions but also reasonable temporal interpolations by explicitly modeling temporal correlation.

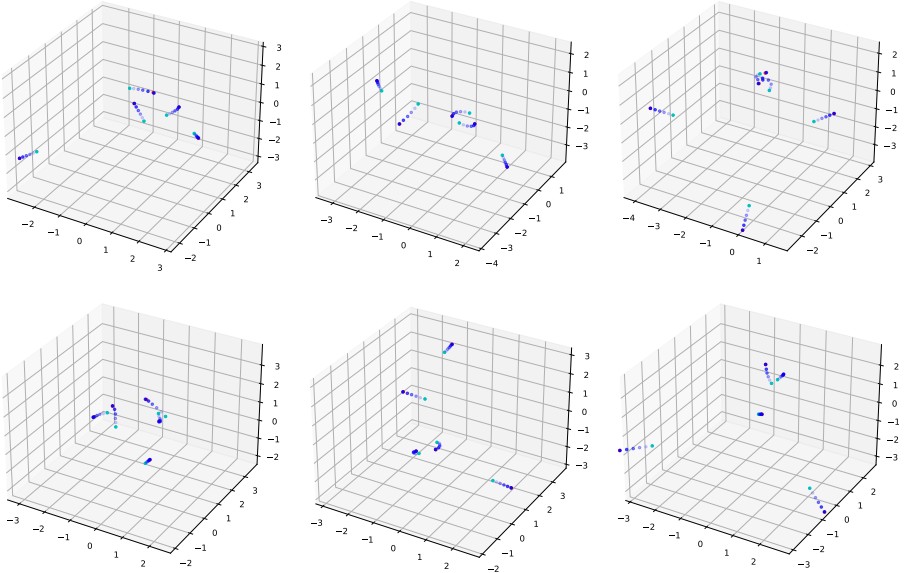

Figure 4: Visualization of the trajectory generated by EGNO with uniform discretization on the N-body simulation dataset. The input is in cyan, the ground truth final snapshot is in red, and the predicted trajectory is in blue. The opacity changes as time elapses.

**Protein.** We provide visualizations of the protein molecular dynamics predicted by EGNO in Figure 7. As shown in the figure, our EGNO can produce accurate final snapshot predictions and also tracks the folding dynamics of the protein. Interesting observations can be found at the bottom regions, *e.g.*, the alpha helix structures, where EGNO gives not only close-fitting predictions but also reasonable temporal interpolations.

### C.4 ZERO-SHOT GENERALIZATION TOWARDS DISCRETIZATION STEPS

We investigate how EGNO generalizes to different discretization steps (*i.e.*, different choices of $P$) in a *zero-shot* manner. To be specific, we load the checkpoint of the trained model with the default discretization steps $P$ shown in Table B.3 and directly conduct inference by setting the number of time steps to $2P$. In particular, for the time embeddings with input $\{\Delta t_1, \cdots, \Delta t_P\}$, we uniformly interpolate in between for the additional time steps, leading to $\{\Delta t_1/2, \Delta t_1, (\Delta t_1 + \Delta t_2)/2, \Delta t_2, \cdots, (\Delta t_{P-1} + \Delta t_P)/2, \Delta t_P\}$, $2P$ points in total. This process increases the temporal resolution without any additional training.

We provide qualitative results in Figure 8. Interestingly, EGNO generalizes to the increased temporal resolution with $2P$ time steps promisingly by yielding accurate and smooth trajectories on top of the low resolution counterparts with $P$ time steps. This capability is remarkably meaningful in the sense that we could potentially train our EGNOs on large scale data with low temporal resolution, and

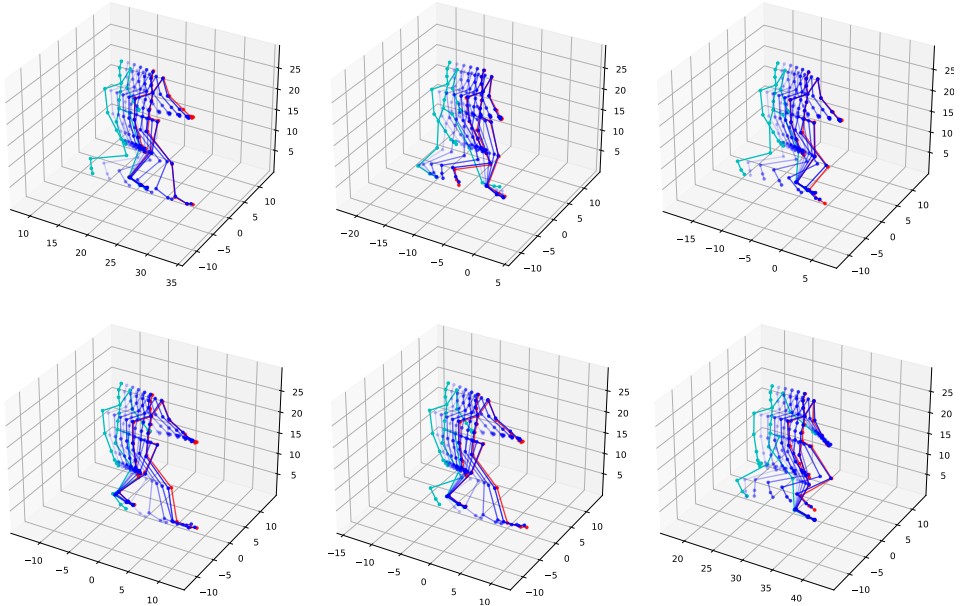

Figure 5: Visualization of the trajectory generated by EGNO with uniform discretization on Motion Capture `Walk`. The input is in cyan, the ground truth final snapshot is in red, and the predicted trajectory is in blue. The opacity changes as time elapses.

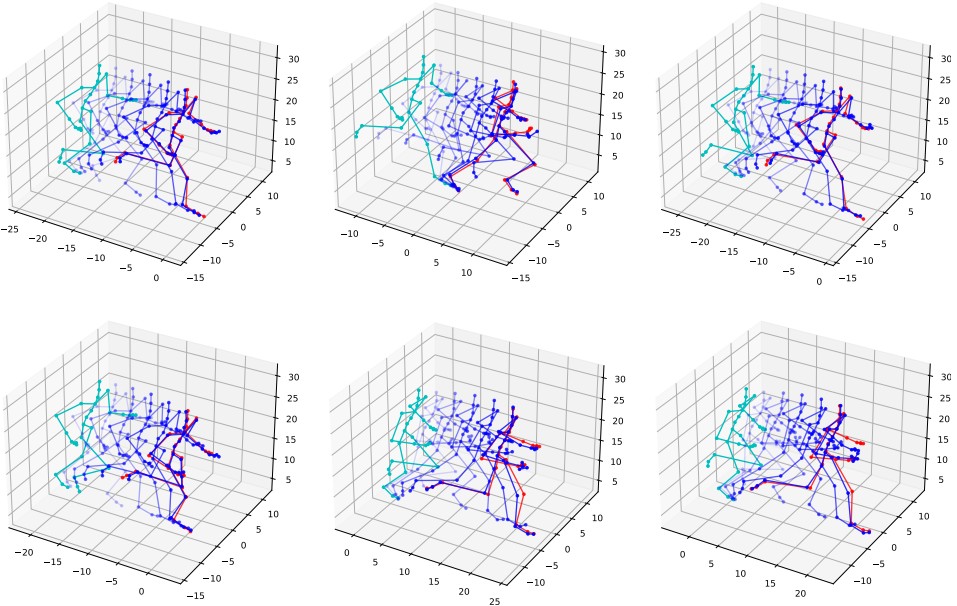

Figure 6: Visualization of the trajectory generated by EGNO with uniform discretization on Motion Capture `Run`. The input is in cyan, the ground truth final snapshot is in red, and the predicted trajectory is in blue. The opacity changes as time elapses.

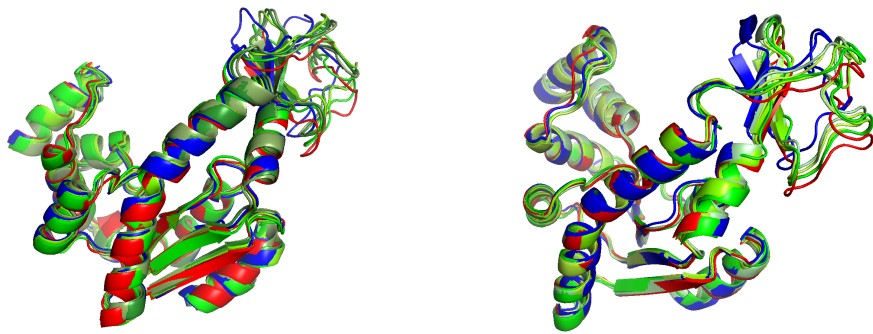

Figure 7: Visualization of the trajectory generated by EGNO with uniform discretization on Protein. The input is in blue, the ground truth final snapshot is in red, and the predicted trajectory is in green. The darkness of green changes as time elapses.

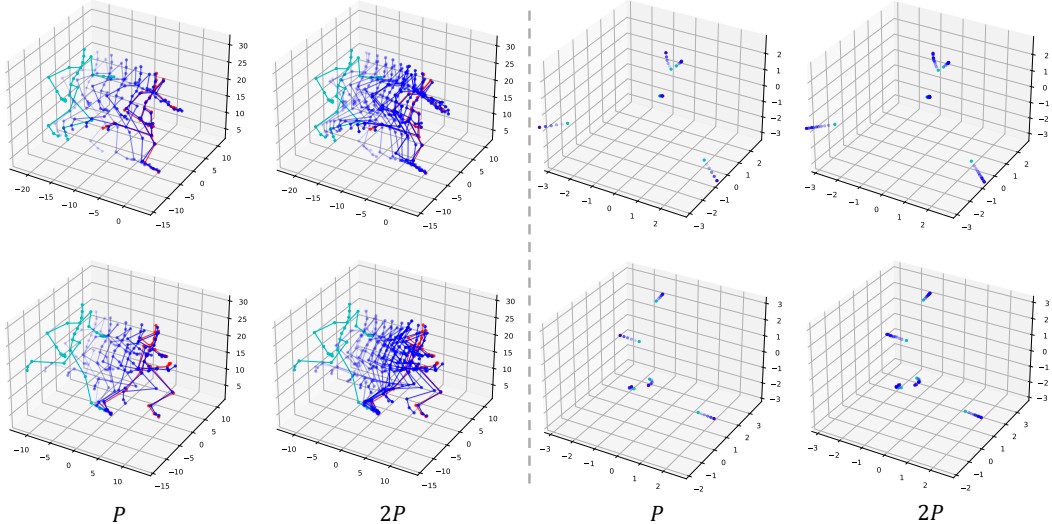

Figure 8: Qualitative comparison of zero-shot generalization towards discretization steps. The sub-figures in columns indexed by $P$ (resp. $2P$) have $P$ (resp. $2P$) time steps. The two sub-figures in the same row share exactly the same initial conditions. Figures on the left are from Motion Capture Run. Figures on the right are from N-body simulation. Better viewed in color.

conduct offline inference per the user's requirement on the temporal resolution. EGNO enjoys the strong generalization we demonstrate here since it directly models the temporal correlations in the frequency domain instead of leveraging temporal operations like rolling out. This observation aligns with the discretization invariance of Fourier Neural Operators (FNOs) manifested in Li et al. (2021).

