# OpenReview forum: "Equivariant Graph Neural Operator for Modeling 3D Dynamics"
_ICLR.cc/2024/Conference — Submitted to ICLR 2024_

### Official Review · Reviewer_kxN5 · 2023-10-29

**Soundness:** 2 fair
**Presentation:** 2 fair
**Contribution:** 2 fair
**Rating:** 5
**Confidence:** 3

**Summary:**

The authors propose a method called Equivariant Graph Neural Operator (EGNO) to directly model dynamics as trajectories instead of just as next-step prediction. To capture the temporal correlations while keeping the intrinsic SE(3)-equivariance, they develop equivariant temporal convolutions parameterized in the Fourier space. Empirical experimental results in multiple domains are provided to support the effectiveness of EGNO.

**Strengths:**

* The paper proposes an effective strategy to capture the temporal correlation along the dynamic trajectory, which results in better prediction accuracy.
* The operator formulation enables efficient parallel decoding of future states within a time window with just one model inference.
* The paper is well-organized and easy to follow.

**Weaknesses:**

* EGNO can be regarded as an extension or application of Fourier operator learning in modeling 3D dynamics. This fact may limit the technical novelty of the paper. Can the authors elaborate on the specific contributions of EGNO beyond the established works in these two fields?
* Some recent related works on N-body simulation tasks, such as GCPNet [1] and ClofNet [2], have not been mentioned or used as baselines in this paper. It would be beneficial to address these works and their relevance to the current research.

[1] https://arxiv.org/pdf/2211.02504.pdf
[2] https://proceedings.mlr.press/v162/du22e/du22e.pdf

**Questions:**

* When it comes to modeling dynamics as a trajectory, another viable approach is to employ neural ODEs as a surrogate for modeling the gradient field between two states. This method can also capture temporal correlation by explicitly reparameterizing an ODE. I noticed that ClofNet utilizes this paradigm to investigate the interpolation and extrapolation capacity of equivariant networks in predicting dynamic trajectories. How does this strategy compare to Fourier operator learning?
* The metrics used for motion capture and molecular dynamics tasks have not been clearly explained. Please provide additional clarification to ensure a better understanding of these aspects.
* The technical details of time embedding have not been thoroughly clarified. It would be helpful to provide more comprehensive explanations.

---

> ### Author Response · Authors · 2023-11-15
> **Response to Reviewer kxN5 Part1**
>
> Thank you for your constructive feedback and questions! The replies to your questions are listed below:
>
> **[W1] EGNO can be regarded as an extension or application of Fourier operator learning in modeling 3D dynamics. This fact may limit the technical novelty of the paper. Can the authors elaborate on the specific contributions of EGNO beyond the established works in these two fields?**
>
> First, we would like to highlight that designing Fourier operators for modeling 3D dynamics is non-trivial: original Fourier operators are designed for scalar functions on the spatial domain, while EGNO is designed for geometric functions on the temporal domain. This requires us to not only model the temporal correlation but also keep the physical symmetries in 3D. We summarize again our technical contributions as follows:
>
> Our work proposes to model the dynamics as trajectories instead of snapshots, which is fundamentally different from previous work.
> We introduce Fourier operators to model the temporal correlations of the dynamics, which is significantly different from previous work where FNOs are typically used for modeling spatial correlations for solving PDEs.
> Most importantly, we design the novel equivariant temporal convolution layer that imposes SE(3)-equivariance in the Fourier layer, where the output trajectory can rotate and translate together with the input trajectory. This again makes our EGNO fundamentally different from existing FNO methods.
>
> **[W2] Some recent related works on N-body simulation tasks, such as GCPNet [1] and ClofNet [2], have not been mentioned or used as baselines in this paper. It would be beneficial to address these works and their relevance to the current research.**
>
> We sincerely thank the reviewer for mentioning these works. GCPNet [1] and ClofNet [2] introduce architectural enhancements with local frame construction to process geometric features while satisfying equivariance, and have been demonstrated to achieve strong performance on N-body simulation. Per the reviewer’s suggestion, we have included the results of GCPNet and ClofNet in Table 1 with the numbers reported in their original papers, and supplement necessary discussions of these works in Related Work.
> In particular, despite using the simple backbone EGNN, our EGNO (F-MSE: 0.0054) still outperforms GCPNet (F-MSE: 0.0070) and ClofNet (F-MSE: 0.0065) with an obvious margin, thanks to the trajectory modeling with equivariant Fourier convolution.
>
> Furthermore, in addition to performance comparison, we want to highlight that these models still focus on modeling the spatial interactions without considering the temporal correlations, while in EGNO, we view dynamics as a function of geometric states over time and concentrate on learning the temporal correlations.

---

> ### Author Response · Authors · 2023-11-15
> **Response to Reviewer kxN5 Part2**
>
> **[Q1] When it comes to modeling dynamics as a trajectory, another viable approach is to employ neural ODEs as a surrogate for modeling the gradient field between two states. This method can also capture temporal correlation by explicitly reparameterizing an ODE. I noticed that ClofNet utilizes this paradigm to investigate the interpolation and extrapolation capacity of equivariant networks in predicting dynamic trajectories. How does this strategy compare to Fourier operator learning?**
>
> ODE and Fourier operator learning can be viewed as two orthogonal methods to model the dynamics. We first present empirical comparisons, and then discuss the methodological differences and connections.
>
> We implement an EGNN trained with ODE objectives named EGNN-ODE, and test the methods by the interpolation (inter) and extrapolation (extra) settings in our N-body simulation task. For EGNO,  we evaluate the interpolation by sampling more timesteps to query the trajectory from the output function, implemented by repeating the input more times and concatenating extra time embeddings specified by the additionally queried intermediate timesteps, similar to our approach in the experiment of Appendix C.4 (C.3 in the original draft).  The extrapolation is evaluated by unrolling the prediction of the dynamics towards the additionally queried timesteps. In this way, both the ODE formulation and our Fourier operator learning approach are capable of handling interpolation and extrapolation scenarios. We report the A-MSE results for both inter and extra trajectories as follows:
>
> | Method | Inter | Extra |
> |  ----  | ---- | ---- |
> | EGNN-ODE | 0.0034 | 0.0620 |
> | EGNO | 0.0022 | 0.0358 |
>
> Empirically, as shown in the table above, the errors of using ODE for both inter and extra are higher than EGNO, due to the error accumulated during solving step by step. By contrast, EGNO yields the prediction over a sequence of structures within the trajectory in parallel with one model call, which mitigates error accumulation in the iterative ODE-solving procedure. Moreover, when conducting the experiment we observed that leveraging ODE indeed introduced significant computation overhead in both training and inference especially when the step size is small, while our EGNO, by modeling the trajectory in the Fourier domain, enables parallel decoding and thus enjoys very high efficiency for both training and inference.
>
> Methodologically, we indeed realize there is even potential to combine ODE and the Fourier neural operator. To be more specific, we can also roll out the dynamics with EGNO for additional timesteps, and utilize the closest predicted point as the initial value to solve the ODE towards other test-time queried time steps. This could potentially offer better prediction accuracy as well as efficiency for ODEs. We leave further studies of the novel combination as future work.
>
> **[Q2] The metrics used for motion capture and molecular dynamics tasks have not been clearly explained. Please provide additional clarification to ensure a better understanding of these aspects.**
>
> Thank you for the advice. The metrics used in motion capture and MD are also F-MSE (Final Mean Squared Error) for the S2S task and A-MSE (Average Mean Squared Error) for the S2T task respectively, exactly the same as those introduced and adopted in the N-body simulation task (c.f. Sec 5.1). We have made this point clearer in the revised manuscript by adding the following statement in Sec. 5.1, right after the introduction of F-MSE and A-MSE: `These metrics are employed throughout the experiments in this paper unless otherwise specified.`
>
>
> **[Q3] The technical details of time embedding have not been thoroughly clarified. It would be helpful to provide more comprehensive explanations.**
>
> Thanks for your suggestion. We added the technical details to the implementation details in the appendix.
>
> We implement the time embedding with sine and cosine functions of different frequencies, following the sinusoidal positional encoding in Transformer [1] (sec 3.5 in the paper). For timestep $\Delta t_i$, the time embedding is implemented by:
>
> $emb_{2j} = sin (i/10000^{2j/d_{emb}})$
> $emb_{2j+1} = cos (i/10000^{2j/d_{emb}})$
>
> where $d_{emb}$ denotes the dimension defined for time embeddings. Actually, the usage of time embedding as well as its implementation is a very common practice in many different domains, like token positional encoding in Large Language Models [1] and denoising time embedding in Diffusion Models [2].
>
> [1] Vaswani, Ashish, et al. "Attention is all you need." Advances in neural information processing systems 30 (2017).
>
> [2] Ho, Jonathan, Ajay Jain, and Pieter Abbeel. "Denoising diffusion probabilistic models." Advances in neural information processing systems 33 (2020): 6840-6851.
>
> ---
>
> We hope our response and the updated paper could address your questions!

---

> ### Author Response · Authors · 2023-11-20
> **Seeking feedbacks during the reviewer-author discussion period**
>
> Dear Reviewer kxN5,
>
> Thank you for your constructive review! This is a kind reminder that as the reviewer-author discussion period is ending soon, we look forward to your feedback of our response to the review.
>
> In our response, we have clarified our technical contributions [W1], added more baselines [W2] and conducted additional experiments [Q1], and further detailed the experimental setup [Q2] and model architecture [Q3].
>
> Thank you again and we sincerely look forward to hearing from you about your feedback to our clarifications!
>
>
> Best,
>
> Authors

---

### Official Review · Reviewer_mHw8 · 2023-10-30

**Soundness:** 3 good
**Presentation:** 3 good
**Contribution:** 3 good
**Rating:** 6
**Confidence:** 3

**Summary:**

This work proposes equivariant temporal convolutions in the Fourier space to capture temporal correlations, which can be observed in complex three-dimensional dynamics. The experimental results show that the method outperforms other equivariant models, such as Tensor Field Networks [Thomas+ 2018], SE(3)-Transformers [Fuchs+ 202], and E(n)-GNN [Satorras+ 2021].

**Strengths:**

* The paper incorporates E(n)-equivariance into Fourier transformation, which is a solid mathematical tool to express temporal dynamics.
* The method is general enough to be compatible with an arbitrary E(n)-equivariant graph neural network.
* The experimental results show that the proposed approach has a significant advantage over other models in tasks of various fields, such as an n-body system, motion capture, and molecular dynamics.
* The paper is well-written enough to understand the methodology and background of the work easily.

**Weaknesses:**

* The method seems to have a strong relation with the temporal bundling proposed in [Brandstetter+, ICLR 2022] in terms of predicting several time steps with one forward computation. Therefore, the reviewer strongly recommends the authors implement the temporal bundling and compare it with the proposed Fourier-based approach to show the superiority.
* The ablation study indicates that two modes are sufficient to perform well on the n-body and mocap-run datasets. However, the advantage of Fourier transformation could be to capture complex temporal dynamics. Therefore, the reviewer suspects that the datasets are too simple to demonstrate the superiority of the method.

Minor points:
* By comparing values, the reviewer guesses Table 3 corresponds to |Train| = 3000 case, but this should be clarified in the caption.

**Questions:**

* The paper states that the authors use fast Fourier transformation (FFT). In my understanding, the length of the time series should be 2^n (n: positive integer) to use FFT, while Table 3 says that they tested time series 2, 5, and 10. How is it possible?
* The experimental conditions are not described in detail enough. P and I are hyperparameters, but the reviewer could not determine the values used for each experiment. Sometimes, one of two parameters is specified, but not two. Thus, could authors clarify these parameters?

---

> ### Author Response · Authors · 2023-11-15
> **Response to Reviewer mHw8 Part1**
>
> Thank you for your constructive feedback and questions! The replies to your questions are listed below:
>
> **[W1] The method seems to have a strong relation with the temporal bundling proposed in [Brandstetter+, ICLR 2022] in terms of predicting several time steps with one forward computation. Therefore, the reviewer strongly recommends the authors implement the temporal bundling and compare it with the proposed Fourier-based approach to show the superiority.**
>
> Thanks for bringing this work to our attention! The temporal bundling (TB) method does have a close relation with EGNO as it is also for the state-to-trajectory task in an orthogonal direction, by factorizing future trajectory. We view TB as a mixture of our designed EGNN-R baseline and EGNO-IV variant in the ablation study of Table 7: EGNN-R learns to predict trajectory by rolling out and EGNO-IV learns to predict trajectory in parallel with a single model call, while TB proposes to factorize trajectory as multiple blocks and learns to predict blocks one by one. We tested such EGNN-TB on the N-body simulation benchmark by factorizing trajectory into blocks of size K=2. The A-MSE result is as follows, which is the accumulated error for unrolling trajectory as the bottom part of Tab.1:
>
> |  | A-MSE |
> |  ----  | ---- |
> | EGNN-TB | 0.0039 |
>
> As shown in the result, EGNN-TB is indeed more expressive than EGNN-R and EGNN-S. However, our method still achieved much better results, which demonstrates the effectiveness of the proposed Fourier method to directly model the temporal correlation over the whole sequence.
>
> However, we want to highlight two points here: 1) [Brandstetter+, ICLR 2022] is related to EGNO on the temporal modeling side. Another key contribution of EGNO is imposing geometric symmetry into the model, making EGNO fundamentally different from [Brandstetter+, ICLR 2022]. 2) Even for temporal modeling, the TB method actually is orthogonal to the Fourier-based method, where we can still factorize predictions as blocks while using Fourier operators to predict each block. Therefore, these two methods are not in conflict and we leave the novel combination as future works.
>
> **[W2] The ablation study indicates that two modes are sufficient to perform well on the n-body and mocap-run datasets. However, the advantage of Fourier transformation could be to capture complex temporal dynamics. Therefore, the reviewer suspects that the datasets are too simple to demonstrate the superiority of the method.**
>
> There are two modes kept for *each of* the feature channels, instead of being only two modes in total. Therefore, it does not mean that the Fourier neural operator can only approximate functions up to 2 modes, and neither can we conclude that the datasets are simple. Indeed, the features in the Fourier domain are pretty high-dimensional and expressive enough to recover complex dynamics. For example, in N-body simulation we have 128 channels for each node, so even for a single node we will keep 128*2 variables in the Fourier space, which is sufficiently expressive to recover complex dynamics.
>
> Actually, keeping limited modes on all channels is a common practice in operator learning works, which has shown effectiveness in a lot of PDE-solving tasks [1]. For example, the classic FNO [1] shows that the neural operator with just 12 modes can already yield significantly better precision than a single function with up to 20 modes (Spectral analysis part in Sec 5.5 [1]).
>
> [1] Li, Zongyi, et al. "Fourier neural operator for parametric partial differential equations."
>
> **[W3] By comparing values, the reviewer guesses Table 3 corresponds to |Train| = 3000 case, but this should be clarified in the caption.**
>
> Thanks for your suggestion, Yes, |Train| = 3000 is our default setting for Table 1 and we followed this setup for Table 3. We have also clarified it in the paper now to make it clearer.
>
> **[Q1] The paper states that the authors use fast Fourier transformation (FFT). In my understanding, the length of the time series should be 2^n (n: positive integer) to use FFT, while Table 3 says that they tested time series 2, 5, and 10. How is it possible?**
>
> Thanks for pointing this out! This is a good point and we will add details to the Appendix.
>
> In this paper, the FFT algorithm is realized by PyTorch implementation. While FFT algorithms are often more efficient when applied to sequences whose lengths are powers of 2, many FFT implementations, including PyTorch implementation, have optimizations that allow them to efficiently process time series of arbitrary lengths.
>
> The requirement for power-of-2 lengths is related to certain specific FFT algorithms rather than a fundamental limitation of the FFT itself. In many cases like PyTorch, the FFT implementation is designed with the Cooley-Tukey FFT Algorithm for General Factorizations, which can handle sequences of arbitrary length, including those that are not powers of 2.

---

> ### Author Response · Authors · 2023-11-15
> **Response to Reviewer mHw8 Part2**
>
> **[Q2] The experimental conditions are not described in detail enough. P and I are hyperparameters, but the reviewer could not determine the values used for each experiment. Sometimes, one of two parameters is specified, but not two. Thus, could authors clarify these parameters?**
>
> Except ablation study, all $I$s are set as default value 2 (already displayed in Table 8 in the Appendix), and the default $P$ is set as 5 for N-body simulation and motion capture, 8 for MD17, and 4 for proteins (clarified in each experiment section). For ablation studies, only the investigated hyperparameter is changed while all other ones are kept as default. We further clarify all $P$ values in corresponding experiment sections and further summarize $I$ values in Table 8 in the Appendix.
>
> ---
>
> We hope our response and the updated paper could address your questions!

---

> > ### Comment · Reviewer_mHw8 · 2023-11-17
> >
> > Thank you for the clarification! I raised the score accordingly (contribution 2 -> 3, overall 5 -> 6).

---

> > > ### Author Response · Authors · 2023-11-18
> > > **Thank you very much for your feedback!**
> > >
> > > Thank you very much for providing valuable feedback and recognizing our efforts!

---

### Official Review · Reviewer_vFqH · 2023-11-03

**Soundness:** 3 good
**Presentation:** 3 good
**Contribution:** 4 excellent
**Rating:** 5
**Confidence:** 4

**Summary:**

The proposed method addresses the limitations of autoregressive models for dynamical systems particularly in settings where graph representations are well motivated with Fourier neural operators. In this setting, roto-translational equivariance is typically incorporated in the graph neural network-based models. This work introduces a temporal convolutional filter that computes the Discrete Fourier transformation while maintaining equivariance to the roto-translational group. This enables the model to combine the strengths of the equivariance graph neural networks and the Fourier neural operators approaches.

The delivered neural operators can produce the evolution of the dynamics of the system (in contrast to the next time step) and they can operate independently of the temporal discretization.

The authors claim that training over temporal dynamics rather than one-step prediction also allows them to achieve superior performance with respect to the MSE on the predictions as well as improvement with respect to the amount of training data required to train the model.

**Strengths:**

The idea of extending equivariant graph neural networks for dynamical systems with neural operators is well-motivated and has clear value.

Theoretically, the paper is well presented. The empirical analysis is very well designed with enough breadth and the results support the proposed claims.

**Weaknesses:**

I have one aspect of the paper that I cannot fully understand and I think the presentation can be improved to benefit a broader set of audience.

The implementation of the method includes the following steps:
- The graph G(t) is repeated P times
- Followed by time embedding of the various length \delta t
- Then a discrete Fourier transformation follows this.

I think the paper can benefit from a more detailed motivation of these steps.

I follow up with specific questions bellow.

**Questions:**

1. What is the role of multiplying the state G(t) of the system?
2. I am not sure how the model does the DFT on a single time set from the input sequence. Can you please explain this?
3. Is this model also able to operate on an input sequence rather than only on one-time step G(t)?
4. Why is there a need to embed the \delta t and how is this implemented in the model?

---

> ### Author Response · Authors · 2023-11-15
> **Response to Reviewer vFqh**
>
> Thank you for your constructive feedback and questions! The replies to your questions are listed below:
>
> **[W] The paper can benefit from a more detailed motivation of these steps.**
>
> Thank you for your suggestions! We also added more details to the paper motivated by your questions. Specific contents are provided below for each question.
>
> **[Q1] What is the role of multiplying the state G(t) of the system?**
>
> The role can be explained from both theoretical and practical views.
>
> From the theoretical view, note that, while EGNO aims for the state-to-trajectory task, we achieve this goal by using neural operators to learn mapping between function spaces. Therefore, for input space, we still need to construct a function $f(t)$ (in practice, a trajectory) that describes structures with time. Repeating $G(t)$ is a simple implementation for this purpose, which constructs $f(t)$ as a constant function.
>
> From the practical view, as discussed in Equation 5, the Fourier layers are implemented with residual connections, which requires the inputs and outputs of each layer to share the same shape. Therefore, we need to repeat $G(t)$ to the same length as the output trajectory.
>
> **[Q2] I am not sure how the model does the DFT on a single time set from the input sequence. Can you please explain this?**
>
> Yes, it would be problematic if we applied DFT just on the repetition of single-time sets. We instead address the problem by expanding the feature of each repeated structure with different time embeddings, which make the sequence of feature mappings distinguishable and thus we can conduct DFT over them to produce meaningful frequency-domain signals.
>
> This in practice is also one empirical reason for incorporating time embeddings, which also partially answers your question Q4.
>
> **[Q3] Is this model also able to operate on an input sequence rather than only on one-time step G(t)?**
>
> Yes. Instead of inputting repetitions of $G(t)$, we can also input a sequence of previous structures to the model, which leads to a trajectory-to-trajectory (T2T) model. We actually have tested this model on N-body simulation before and report the result as follows, named EGNO-T2T. The reported number is A-MSE, the same as the bottom part in Tab.1.
>
> |  | A-MSE |
> |  ----  | ----  |
> | EGNO-T2T | 0.0020 |
>
> As shown in the result, our EGNO can handle the T2T task and even achieve better results. However, we emphasize that, although this implementation achieved better results than the original EGNO, we don’t want to highlight this because EGNO and all baseline models only operate on a single input point and EGNO-T2T makes use of more input information. Therefore, we take this just as an ablation study and leave detailed investigations of the T2T framework as a promising future direction.
>
> **[Q4] Why is there a need to embed the \delta t and how is this implemented in the model?**
>
> One partial reason is given in our response to your Q2. More explanations are given as follows:
>
> As explained in our response to Q1, repetitions of $G(t)$ will be mapped to the output function describing the predicted trajectory. Therefore, we need to make the model aware of the timestep information that we are interested in for predicting future trajectory. Indeed, the usage of time embedding as well as its implementation is a very common practice in many different domains, like token positional encoding in Large Language Models [1] and denoising time embedding in Diffusion Models [2].
>
> We implement the time embedding with sine and cosine functions of different frequencies, following the sinusoidal positional encoding in Transformer [1] (Sec 3.5 in the paper). For timestep $\Delta t_i$, the time embedding is implemented by:
>
> $emb_{2j} = sin (i/10000^{2j/d_{emb}})$
> $emb_{2j+1} = cos (i/10000^{2j/d_{emb}})$
>
> where $d_{emb}$ denotes the dimension defined for time embedding, which is set as 32 in all our experiments (see Table 8).
>
> We added the details to the implementation details in the appendix.
>
> [1] Vaswani, Ashish, et al. "Attention is all you need." Advances in neural information processing systems 30 (2017).
>
> [2] Ho, Jonathan, Ajay Jain, and Pieter Abbeel. "Denoising diffusion probabilistic models." Advances in neural information processing systems 33 (2020): 6840-6851.
>
> ---
>
> We hope our response and the updated paper could address your questions!

---

> > ### Comment · Reviewer_vFqH · 2023-11-21
> > **Clarification on Q2**
> >
> > Thanks for all the answers. Your responses cleared a number of points.
> >
> > I do, however, have a follow-up on question 2. Your answer indicates that DFT only makes sense on the input because you have added the time embeddings and hence something is changing over time. However, this is not the actual signal as the time embeddings do not carry information about the system's evolution up to that point. So, I am unsure of the value of operating in the frequency domain. Can you please explain, what is the added value of operating in the frequency domain when your model is processing one system state as input?

---

> ### Author Response · Authors · 2023-11-20
> **Seeking feedbacks during the reviewer-author discussion period**
>
> Dear Reviewer vFqH,
>
> Thank you for your review, helpful suggestions, and recognition of the presentation and contribution of our paper!
>
> This is a kind reminder that as the reviewer-author discussion period is ending soon, we look forward to hearing from you about your feedback to our response.
>
> In particular, we have added more details on the motivations of the designs, per your advice in the review. Please refer to our posted point-to-point response.
>
> Thank you again and we sincerely look forward to your feedback!
>
> Best,
>
> Authors

---

> ### Author Response · Authors · 2023-11-21
> **Further explanation on Q2**
>
> Dear Reviewer vFqH,
>
> Thank you for the response and follow-up. We are very willing to further clarify this point in detail.
>
> *The temporal signal is not simple and it is meaningful to project it to the frequency domain.* Although the model seems to take in only one system state as input, with the help of the time embedding, after the first Fourier layer the model indeed produces an estimated trajectory of the system states, where the states at different time steps are no longer the same. Such temporal convolution is repeated several times (by stacking several such layers), until at the final output the trajectory is supposed to match the given target trajectory, which is a very complex state signal along the time dimension. Therefore, the input signal for the Fourier convolutions is getting more complex as it is processed by multiple such layers, even as complex as the target trajectory at the final layer, which is the output of the entire model. In consequence, learning in the frequency domain is meaningful.
>
> Note that the interesting property of transforming simple temporal functions to complex temporal functions is fulfilled by the learnable parameters placed in the frequency domain, which are optimized during training.
>
> We also notice that the reviewer raises the point that `However, this is not the actual signal as the time embeddings do not carry information about the system's evolution up to that point.` While it seems that the input time embeddings do not carry information about the system's evolution, it is not the case since the time embeddings, after transformed by DFT, are used to index the learned parameters in frequency domain, which contain rich information on how to transform the input repetitions of states into the output predicted trajectory. That is, instead of only considering the input information, it would be more reasonable to consider the posterior induced by both the input and the learned parameters $\theta$ of the model, since the parameters are learned under the supervision of the target trajectory, which are consequently very important and encodes rich temporal correlations.
>
> *Leveraging Fourier convolution improves the performance.* The empirical advantage of using temporal convolution parameterized in frequency domain is also advocated by the experiments in ablation studies (Table 7), where our full version of learning $h,x,v$ all in frequency domain significantly improves the performance over the variant IV without any parameterization in frequency domain. We provide simplified results in the table below, and encourage the reviewer to refer to Table 7 in the manuscript for full results.
>
> | Method | N-body | Mocap-Run |
> |  ----  | ---- | ---- |
> | EGNO w/o frequency domain | 0.0072 | 48.2 |
> | EGNO | 0.0055 | 33.9 |
>
>
> *Learning in frequency domain permits zero-shot generalization towards different discretization steps.* Another benefit of such parameterization in frequency domain is that the model can generalize to different granularities of the temporal discretization, since we place the learned parameters in the frequency domain and leverage DFT/IDFT as the change-of-basis. We have demonstrated such interesting effect in Appendix C.4.
>
> Thank you again and we are very willing to discuss more if there are still any questions.
>
> Best,
>
> Authors

---

### Official Review · Reviewer_AtkK · 2023-11-05

**Soundness:** 3 good
**Presentation:** 3 good
**Contribution:** 3 good
**Rating:** 8
**Confidence:** 3

**Summary:**

This paper proposes a method to learn SE(3) equivariant discretization invariant dynamics on graphs. To do this, the authors learn a Fourier neural operator (FNO) to predict future states from the current state. Differently from previous approaches, here, future states are determined by a time window (making it a function space), rather than a fixed dt. To constrain the dynamics to be equivariant to SE(3) transformations, the authors use a modified version of the FNO layer.

**Strengths:**

*Originality:* The work appears original.
- The literature review appears comprehensive
- The work seems to address a gap in the literature

*Quality:* The work is of good quality.
- The claims are validated either theoretically or empirically.
- In the many experiments presented, the proposed work outperforms baselines
- The ablation studies show the utility of each component of the approach
- The empirical analysis studies the effect of varying I and P and finds results consistent with expectations

*Clarity:*
- The work is clearly written and organized.

*Significance:* The work appears significant.
- As far as I know this work is the first to consider temporal windows as function spaces on which to apply neural operators.

**Weaknesses:**

No notable weaknesses

**Questions:**

In the paper, $M_\theta$ is defined as a complex tensor, is this reflected in the implementation? How does that impact computational performance?

---

> ### Author Response · Authors · 2023-11-15
> **Response to Reviewer AtkK**
>
> Thank you for your question!
>
> **[Q] In the paper, $M_\theta$ is defined as a complex tensor, is this reflected in the implementation? How does that impact computational performance?**
>
> Yes, this complex tensor is reflected in our implementation. We implement the complex tensor with two tensors to represent the real and imaginary components of complex numbers, and use `torch.view_as_complex` to convert the two tensors into the complex one. The computation is roughly as efficient as typical real tensors.
>
> We add this information to the implementation details in the appendix.
>
> ---
>
> We hope our response and the updated paper could address your questions!

---

### Author Response · Authors · 2023-11-15
**General response to all reviewers**

We would like to thank all the reviewers for your constructive comments and suggestions!

We replied to the questions individually and added additional content to the paper according to your suggestions, including **extra experiments, more implementation details, and related works**. Due to the page limit, most of them are updated to the Appendix. The updated text is colored in purple.

We hope our response and the updated paper could address your questions!

---

### Meta-Review · Area_Chair_LT8P · 2023-12-09

**Metareview:**

This paper studies 3D dynamics using graph neural operators. While the application of neural operators on 3D dynamics is interesting, the reviewers are concerned with the technical advances of this work as it represents an extension and use of current methods for an interesting application. Given the nature of ICLR, I lean towards rejecting this paper.

**Justification For Why Not Higher Score:**

While the application of neural operators on 3D dynamics is interesting, the reviewers are concerned with the technical advances of this work as it represents an extension and use of current methods for an interesting application.

**Justification For Why Not Lower Score:**

NA

---

### Decision · Program_Chairs · 2024-01-16

Reject